# Demonstration of CMOS-compatible memristor-based electrochemical biosensor transducer with threshold-sensing functionality

Young-Joon Kim [1,2,9] ✉, Youna Kwon[3,9], Youngwoo Yoo [1,2,9], Kandaswamy Theyagarajan [1,2], Sairaman Saikrithika[1,2], Aryeong Lee[4], Nam Ho Bae[3], Won-Chul Lee[3,5], Gapseop Sim[3,5], Younghyun Lee [6], See-On Park [7], Hyijae Cho[3,5], Min-Ho Kang[3], Youngjoo Kim[3], Yumin Park[3], Kyoung G. Lee [3], Choul-Young Kim[5], Hyoungho Ko[5], Woo-Suk Sul[3], Seok Jae Lee[3], Jae-Hyuk Ahn[5], Shinhyun Choi [7] ✉, Kyung Min Kim [6] ✉ & Jongwon Lee [8] ✉

Many electrochemical biosensors operate based on a threshold-sensing (TS) method, which indicates the presence of a disease when the concentration of a biomarker exceeds a predetermined, disease-specific threshold. The TS in biosensor systems is often implemented using power-hungry signal processing (SP) modules or computers, which increases energy consumption and system complexity. Here, we propose a memristor-based bio-to-electrical transducer with built-in TS functionality, allowing TS to be performed directly within the transducer instead of relying on SP modules and computers. Fabricated resistive random-access memory-based $TaO_x/Ta_2O_5$ memristors meet the transducer requirements, such as a high on/off ratio greater than 30 while maintaining a long unit pulse width exceeding 10 µs. The intended operation of the proposed transducer was experimentally confirmed by the immediate change in resistance of the memristor ($R_M$) from high resistance state to low resistance state. Using this proposed transducer, a complete electrochemical biosensor system was implemented by integrating a sensor electrode for pH sensing, an SP module, and a display with the proposed transducer. The memristor-based system offers flexible control of the threshold pH point through a simple design, making it well-suited for point-of-care diagnostics, where portability is highly essential.

The growing demand for portable bio-healthcare diagnostic devices is driven by the need for high energy efficiency, compact chip design, and efficient computation tailored to the diagnosis of various diseases. This increasing interest is further fueled by an aging population and the rising need for real-time diagnostic capabilities that are not constrained by time or location. Electrochemical biosensor systems, among the most prominent portable diagnostic devices, have been actively researched using various semiconductor devices and

nanomaterials to detect serious diseases, such as influenza, diabetes, cancer, cardiovascular conditions, and digestive disorders[1–5]. The biosensor systems generally consist of analytes, transducers, signal processing modules (SPs) like readout integrated circuits (ICs) and digital signal processing (DSP), and displays. These systems operate by converting the biochemical signals from the analytes into electrical signals through transducers, which are then either processed by the SPs within the devices or transmitted wirelessly via Bluetooth to a computer for processing by software programs[6,7]. The processed signals are compared to unique reference values for each disease, allowing the system to determine whether the disease is present or to assess the level of risk, which is then communicated via the display.

Many electrochemical biosensors operate based on a threshold-sensing (TS) method, which determines the presence of a disease when the concentration or amount of a biomarker in the analyte exceeds a predetermined, disease-specific threshold[8–10]. For example, COVID-19 has key biomarker thresholds, such as lactate dehydrogenase < 365 U/L, lymphocytes > 14.7%, and high-sensitivity C-reactive protein < 41.2 mg/L[8]. The diagnostic efficiency of acute myocardial infarction is improved when the troponin T concentration is higher than 1 ng/mL[9,10]. Efficient implementation of the TS function in biosensor systems is of critical importance. Typically, TS is performed in the SP module or an external computer; however, this approach inevitably increases power (or energy) consumption and system complexity due to the reliance on power-hungry ICs, such as analog-to-digital converters, comparators, digital signal processors (DSPs), and computer. If the TS function can be embedded directly into the transducer itself rather than in the SP or computer, a more scalable and efficient biosensor system can be achieved. To date, there have been no reports on bio-to-electrical transducers with the TS function.

A memristor device with a simple structure of metal-insulator-metal (MIM) is a next-generation memory device with non-volatility and variable resistance characteristics, using a mechanism of production and rupture for conductive filaments due to the intrinsic redox reaction[11]. Owing to the dense cell size of $4F^2$ and low set/reset voltage of less than 1 V of the memristors, they have been used as cells in cross-bar arrays of neuromorphic systems and have reduced power consumption and improved energy efficiency of the system[12,13]. For a neuromorphic system to reach levels comparable to the human brain, which consumes 20 W of power, the memristor devices must ultimately be able to consume less than 10 fJ per synaptic operation and more research is underway to reduce operating current, voltage and latency while having the compatibility with CMOS process, which is essential to be co-integrated with peripheral circuits in the systems[14–16]. Meanwhile, the non-volatility and variable resistance characteristics of the memristor allow it to be used as a bio-to-electrical transducer in the electrochemical biosensor system[17–19]. Ref. 17 exposed an analyte solution to an insulator region of an ITO/TiO2/Al MIM memristor and showed a change in the device's on/off resistance ratio in response to variations in the concentration of the biomarker, nonstructural protein 1 (NS1 protein). When the well diameter in the insulator region exceeded 1.5 mm, a fairly linear relationship between NS1 protein concentration and the on/off resistance ratio characteristics was observed. Ref. 18 exposed a PSA analyte to the Pt metal region of a Pt/Al2O3/TiO2/Pt memristor, where the resistance value of the device itself changed in response to the concentration of the biomarker, although the relationship was not linear. These previous results were iterative implementations for the conventional bio-to-electrical transducers without any TS functionality and revealed the limitations to the reusability of biosensors by directly exposing the memristors built with expensive semiconductor processes to the analyte solutions[17–19]. Furthermore, prior studies have often assumed the availability of precise pulse generation, a requirement that may pose challenges for low-cost home-diagnostic devices employing crystal-less

microcontroller units[20]. Also, the stochastic nature of resistive random-access memory (RRAM) can lead to inaccurate readings and compromised diagnostic results in actual deployments[21].

To address the above-mentioned constraints, we propose a memristor-based bio-to-electrical transducer with integrated TS functionality, designed to ensure reusability for electrochemical bio-sensor applications. Here, RRAM-based TaO$_X$/Ta2O5 memristor devices were fabricated using a stable 8-inch Si CMOS process line. By controlling an oxygen reservoir layer of TaO$_X$, a memristor species were tailored to meet the proposed transducer requirements, including high on/off ratio and stable operation under long pulse widths. By applying the developed memristor, a complete electrochemical bio-sensor system was implemented by integrating a sensor electrode, an SP module, and a display with the proposed transducer. The intended TS function of the proposed transducer was experimentally demonstrated by confirming that the resistance of the memristor ($R_M$) changed from high resistance state (HRS) to low resistance state (LRS) at a specific value of pH solution (named threshold pH point), and thereby the output voltage changed immediately from low to high values. The threshold pH point (TP) of the system can be tuned to any desired pH level by adjusting the integrated reference voltage ($V_{REF}$), demonstrating that the system is capable of not only diagnosing diseases but also providing prognostic insights. Using this approach, a simplified biosensor system was developed by replacing bulky components with compact ICs. Therefore, the proposed memristor-based electrochemical transducer technology represents a promising candidate for highly portable point-of-care diagnostic devices.

## Results
### Proposed memristor-based electrochemical biosensor transducer

Figure 1a shows a circuit diagram of the proposed bio-to-electrical transducer with TS functionality. The transducer consists of a sensor electrode, two FETs, a memristor with a resistance of $R_M$, and an output resistor with a resistance of $R_O$. A gate terminal of an upper FET is extended and connected to the sensor electrode. An analyte solution is dropped onto the extended gate region (i.e., the sensor electrode), not on the memristor, so that the reusability issue of the previous memristor-based transducers is addressed[17–19]. The lower FET provides a variable resistor with a resistance of $R_T$ by controlling a gate voltage ($V_{GS}$). While previous studies have used memristors with multi-bit characteristics[17–19], this work utilizes a single-bit memristor with only two resistances: high resistance state ($R_{HRS}$) and low resistance state ($R_{LRS}$), as shown in Fig. 1b, which enables the transducer to exhibit TS functionality. The single-bit memristor is characterized by the change of $R_M$ from $R_{HRS}$ (or $R_{LRS}$) to $R_{LRS}$ (or $R_{HRS}$) through the SET (or RESET) process by applying a set voltage ($V_{SET}$) (or reset voltage ($V_{RST}$)), while a forming process with a forming voltage ($V_{FORM}$) is required to initially form conductive filaments of the memristor[22–24].

The key operation of the proposed transducer with TS functionality is set up with its timing diagram in Fig. 1c, which consists of three stages: initialization, sensing, and reading. In the initialization stage, $R_M$ is set to $R_{HRS}$ by the forming and reset processes of the memristor. When an analyte solution with a biomarker concentration of $C$ is introduced and a reference voltage ($V_{REF}$) is applied, the surface potential of the gate electrode ($V_W$) changes due to an intrinsic chemical reaction. At the sensing stage, $V_{GS}$ and $V_D$ are applied. The change in $V_W$ modulate the drain current ($I_D$) according to the current–voltage characteristics of the upper FET. The output voltage ($V_{OUT}$) is formed as the product of $I_D$ and the equivalent value ($R_{EQ}$) of the bottom three bottom resistors with $R_T$, $R_M$, and $R_O$. Because $R_T$ is designed to be considerably lower than $R_M$ and $R_O$, $V_{OUT}$ is approximately given by $I_D \times R_T$, and is almost independent of changes in $R_M$, as seen in Fig. 1d. When $V_{OUT}$ exceeds $V_{SET}$ of the memristor, $R_M$ abruptly changes from $R_{HRS}$ to $R_{LRS}$ if $C$ exceeds the threshold concentration

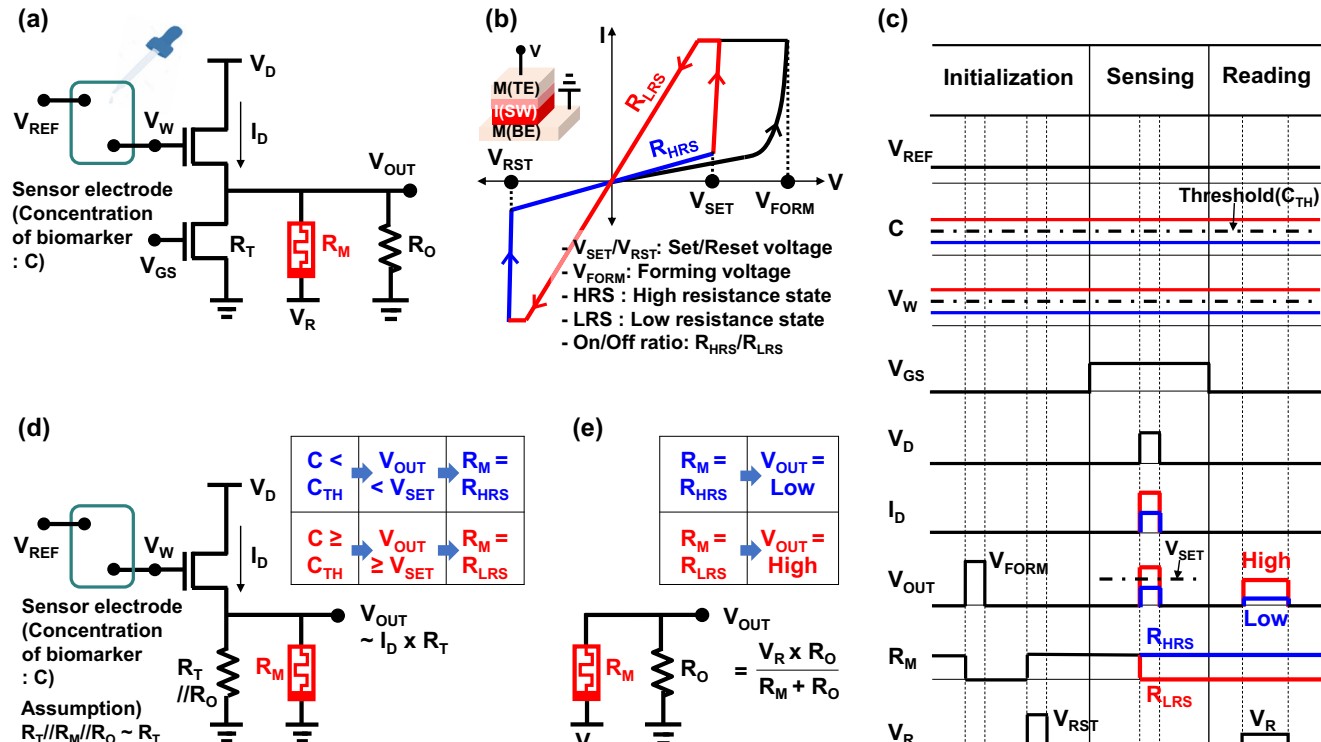

**Fig. 1 | Circuit diagram and timing diagram of the proposed memristor-based bio-to-electrical transducer with threshold-sensing functionality. a** A circuit diagram of the transducer, consisting of a sensor electrode, two field effect transistors (FETs), a memristor with a resistance of $R_M$, and an output resistor with a resistance of $R_O$. A gate terminal of an upper FET is extended to the sensor electrode and exposed to an analyte solution. A lower FET provides a variable resistor with a resistance of $R_T$. **b** An operation of a memristor with a single-bit property. A set voltage ($V_{SET}$) (or reset voltage ($V_{RST}$)) is defined as the voltage applied for the set process (or reset process), changing from the high resistance state (HRS) (or low resistance state (LRS)) to LRS (or HRS). A forming voltage ($V_{FORM}$) refers to the voltage applied during the forming process that initially forms conductive filaments in the memristor. An on/off ratio is defined as the ratio between the resistance of HRS ($R_{HRS}$) and the resistance of LRS ($R_{LRS}$). **c** A timing diagram of the proposed memristor-based transducer, consisting of initialization, sensing, and reading stages. At the initialization stage, the forming and reset processes of the memristor are performed so that the $R_M$ value becomes $R_{HRS}$. At the same time, an analyte solution with a biomarker concentration of $C$ drops to the sensor electrode, and a reference voltage ($V_{REF}$) is applied. As $C$ increases, the surface potential of the extended gate ($V_W$) increases compared to $V_{REF}$. At the sensing stage, a gate voltage of the lower FET ($V_{GS}$) and a drain voltage ($V_D$) are applied. The upper FET converts from the $C$ to a drain current ($I_D$). An output voltage ($V_{OUT}$) is formed as the product of $I_D$ and an equivalent resistance of $R_T$, $R_M$, and $R_O$. When $V_{OUT}$ exceeds $V_{SET}$, $R_M$ is converted from $R_{HRS}$ to $R_{LRS}$. At the reading stage, $V_{GS}$ and $V_D$ are grounded to turn off the two FETs, and a reading voltage ($V_R$) is applied. Consequently, $V_{OUT}$ exhibits either a high or low level, corresponding to $R_{LRS}$ or $R_{HRS}$. **d** An equivalent circuit of the transducer at the sensing stage. **e** An equivalent circuit of the transducer at the reading stage.

($C_{TH}$) (Fig. 1c, d). Here, the turn-on time of $V_D$ determines the sensing duration and is made as short as possible to minimize stress on the memristor. In the reading stage, $V_{GS}$ and $V_D$ are grounded to turn off the two FETs, and a reading voltage ($V_R$) is applied. After turning off the two FETs, a voltage divider circuit is formed by $R_M$, $R_O$, and $V_R$, as shown in Fig. 1e. Based on an inherent equation of $V_{OUT} = V_R \times R_M/(R_M + R_O)$, $V_{OUT}$ exhibits a threshold function of high or low output levels corresponding to $R_{LRS}$ and $R_{HRS}$, respectively. Although there is a time interval between the sensing and reading stages, the unique memory function of the memristor allows the $R_M$ value obtained in the sensing stage to be effectively utilized in the reading stage without any loss of resistance values.

It is paramount to develop a memristor customized for the transducer with TS functionality. The transducer's TS function is successfully conducted when the high $V_{OUT}$ is at least 0.1 V higher than the low $V_{OUT}$. Since we aimed to implement a proof-of-concept transducer prototype that integrates a fabricated memristor through wire bonding and does not include ESD protection techniques, the performance degradation caused by wire bonding and the absence of ESD protection should be considered when defining the memristor specifications[22]. Accordingly, under the condition that $V_R$ and $R_O$ were 0.3 V and 560 Ω, respectively, the on/off ratio (i.e., the ratio of $R_{HRS}$ and $R_{LRS}$) of the memristor that satisfies the $V_{OUT}$ difference of 0.1 V was

targeted as 30. In developing the transducer, real-time monitoring of the TS performance is crucial, necessitating the display of values on the $V_{OUT}$ node in real-time. To achieve this, a commercial ADC with a sampling rate of 200 kilo samples per second (kS/s) was interfaced between the $V_{OUT}$ node and the display. Consequently, voltage pulses with a minimum duration of 10 μs, corresponding to the sampling rate, were applied to the $V_{OUT}$ node, which required a reliable memristor capable of withstanding such extended pulse durations. Moreover, for the envisioned mass production of a self-diagnostic device, although monitoring $V_{OUT}$ is not required for data extraction, generating precise short-duration High pulses (<10 μs) in a cost-effective, crystal-less manner remains a challenge. Thus, our study presents a high-performance memristor device specifically engineered to achieve an on/off ratio surpassing 30, even when subjected to pulse durations exceeding 10 μs. This optimization renders the device well-suited for deployment in bio-to-electrical transducers, ensuring reliable performance in real-world applications.

## CMOS process-compatible TaO$_X$/Ta$_2$O$_5$ memristors for the bio-sensor transducers

A switching layer (SW) stack of TaO$_X$/Ta$_2$O$_5$ has been largely utilized as memristors[23,25–27], because of their stable switching operation by means of low Gibbs-energy-based redox reactions[28] and excellent

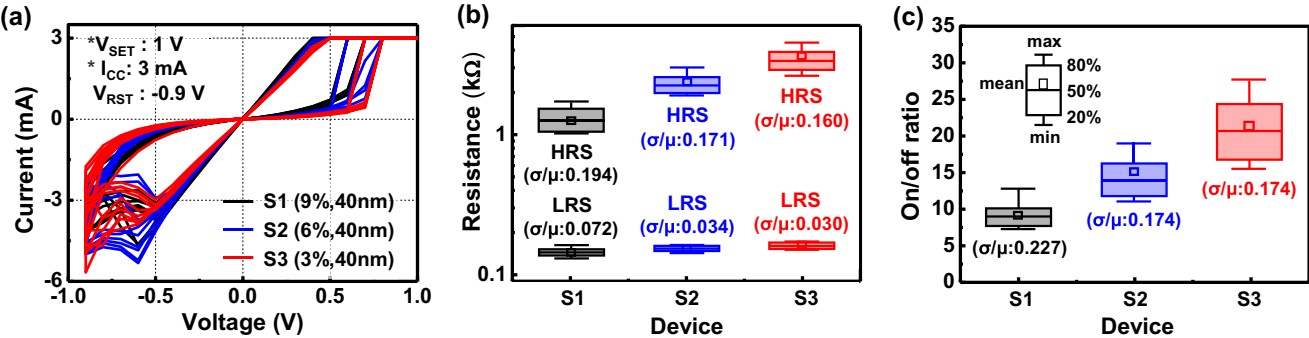

**Fig. 2 | Measured electrical characteristics of CMOS process-compatible TaO_X/Ta_2O_5 memristors with different O_2/Ar gas mixture ratios in TaO_X.**
**a–c** Measured I–V curves, LRS/HRS resistances, and on/off ratios of the memristors based on 40 nm-thick TaO_X layers with O_2/Ar gas mixture ratios of 9 (S1), 6 (S2), and 3% (S3). The thickness of Ta_2O_5 layers in S1–S3 species is 4 nm. For each species, ten devices were randomly selected on an 8-inch wafer. LRS/HRS resistances were measured at a read voltage of 0.3 V.

**Table 1 | Conceptual switching mechanisms of CMOS process-compatible TaO_X/Ta_2O_5 memristors with different O_2/Ar gas mixture ratios in TaO_X**

| TaO_X O_2/Ar ratio | TaO_X X mole fraction | Ta_2O_5 O/Ta ratio | V_FORM_mean (V) | I_FORM_mean (mA) | Filament generation | R_LRS_mean (Ω) | Filament Rupture | R_HRS_mean (Ω) | On/off Ratio_mean |
|---|---|---|---|---|---|---|---|---|---|
| 9% (S1) | 1.04 (See Fig. S1d) | 2.0 (See Fig. S1d) | 2.16 (See Fig. S2a,d) | 2.44 (See Figs. S2a,e) | Initial → Forming & Set | 141 (See Fig. 2b) | Reset | 1231 (See Fig. 2b) | 8.8 (See Fig. 2c) |
| 6% (S2) | 0.54 (See Fig. S1e) | 1.25 (See Fig. S1e) | 1.66 (See Figs. S2b,d) | 2.20 (See Figs. S2b,e) |  | 151 (See Fig. 2b) |  | 2276 (See Fig. 2b) | 15.1 (See Fig. 2c) |
| 3% (S3) | 0.39 (See Fig. S1f) | 0.71 (See Fig. S1f) | 1.34 (See Figs. S2c,d) | 2.07 (See Figs. S2c,e) | ○: oxygen vacancy, ●: oxygen ions | 158 (See Fig. 2b) | ○: filament rupture | 3401 (See Fig. 2b) | 21.6 (See Fig. 2c) |

CMOS process compatibility[23,25–27]. TaO_X/Ta_2O_5 memristors were fabricated on 8-inch Si wafers, with an SW stack having a reactive sputtered 40 nm-thick TaO_X layer as an oxygen reservoir and a 4 nm-thick atomic layer deposition (ALD)-deposited Ta_2O_5 layer. The top (TE) and bottom (BE) electrodes of the two species were made of TiN metal with a thickness of more than 100 nm, and their device fabrication is detailed in the "Methods" section. The O_2/Ar gas mixture ratio of the 40 nm-thick TaO_X layer was varied with 9 (S1), 6 (S2), and 3% (S3), which corresponds to the O_2/Ar gas amounts with 3/30.3 sccm (S1) to 2/31.3 sccm (S2), and 1/32.3 sccm (S3), respectively. As determined from measured electrical results shown in Fig. 2a–c, the S1, S2, and S3 species exhibited average $R_{LRS}$ values of 141, 151, and 158 Ω; average $R_{HRS}$ values of 1231, 2276, and 3401 Ω; and corresponding average on/off ratios of 8.8, 15.1, and 21.6, respectively. As the O_2/Ar gas mixture ratio in the TaO_X layer decreased from 9 to 3%, the $R_{LRS}$ slightly increased, and the $R_{HRS}$ significantly increased, resulting in an increase in the on/off ratio from 8.8 to 21.6. This increase in on/off ratio is attributed to the decrease in the size of the filament, as seen in the proposed current switching mechanism (Table 1). From EDS scan results of fully fabricated memristors (Supplementary Fig. S1), TaO_X layers of S1 and S3 exhibited O/Ta ratios of 1.04 and 0.39, respectively, while the corresponding Ta_2O_5 layers showed 2 and 0.71, which indicate that the TaO_X/Ta_2O_5 SW stack of S3 was more oxygen-deficient than that of S1. As the oxygen-deficient characteristic becomes stronger (i.e., Ta becomes richer), the electrical resistance of the SW stack itself decreases, which lowers the breakdown voltage for filament formation and results in thinner filaments[28]. The reduction in forming voltage ($V_{FORM\_mean}$) and current ($I_{FORM\_mean}$) of S3 compared to ones of S1 (Supplementary Fig. S2) is a piece of evidence supporting the generation of the thinner filaments. The thin filaments undergo

**Table 2 | Conceptual switching mechanisms of CMOS process-compatible TaO$_X$/Ta$_2$O$_5$ memristors with different TaO$_X$ thicknesses**

| TaO$_X$ Thick-ness | TaO$_X$ X mole fraction | TaO$_X$ Resistance (mΩ) | I$_{FORM}$_mean (mA) | Filament generation | | R$_{LRS}$_mean (Ω) | Filament Rupture | R$_{HRS}$_mean (Ω) | On/off Ratio _mean |
|---|---|---|---|---|---|---|---|---|---|
| | | | | Initial | Forming & Set | | Reset | | |
| 40 nm (S2) | 0.54 (See Fig. S4b) | 36 (See Fig. S4a) | 2.20 | | | 151 (See Fig. 3b) | | 2276 (See Fig. 3b) | 15.1 (See Fig. 3c) |
| 20 nm (S5) | 0.54 (See Fig. S4c) | 19 (See Fig. S4a) | 1.6 | | ○: oxygen vacancy, ●: oxygen ions | 185 (See Fig. 3b) | ○: filament rupture | 6525 (See Fig. 3b) | 35.2 (See Fig. 3c) |

stronger filament rupture than the thick filaments[29], assuming that the same number of oxygen anions contribute to filament rupture, as shown in the filament rupture (Reset) column in Table 1. The stronger filament rupture means higher $R_{HRS}$, leading to a higher on/off ratio. As all fabricated memristors exhibited Schottky emission characteristics in voltage ranges from 0.15 to 0.55 V in the HRS, as described by measured logarithmic I–V curves for the conduction mechanism in Supplementary Fig. S3, it can be inferred that the filament rupture initiates near the TaO$_X$/Ta$_2$O$_5$ interface, where a Schottky barrier exists between the two layers.

The TaO$_X$ thickness was reduced from 40 to 20 nm while maintaining the thickness of Ta$_2$O$_5$ at 4 nm. The O$_2$/Ar gas mixture ratio of TaO$_X$ was varied at 9 (S4), 6 (S5), and 3% (S6). Measured $R_{LRS}$/$R_{HRS}$ values for S4, S5, and S6 were 160/3133, 185/6525, and 224/3364 Ω on average, and their on/off ratio values were 19.7, 35.2, and 15.6 on average, respectively (Supplementary Fig. 3a–c). The on/off ratio values (19.7 and 35.2) obtained at O$_2$/Ar gas mixture ratios of 9 and 6% were more than twice those (8.8 and 15.1) of 40 nm-thick TaO$_X$-based samples. This on/off ratio increase is also attributed to the decrease in filament size, as shown in the proposed current switching mechanism in Table 2. The electrical resistance of 20 nm-thick TaO$_X$ layers was smaller compared to the 40 nm-thick species due to their less thickness. For example, in the case of TaO$_X$ layers with an O$_2$/Ar gas mixture ratio of 6%, 20 nm-thick and 40 nm-thick TaO$_X$ layers showed average resistances of 19 and 36 mΩ (see Supplementary Fig. S4a), respectively, while achieving the same x mole fraction value of TaO$_X$ of 0.54 (see Supplementary Fig. S4b, c). The reduced resistance results in a smaller soft breakdown voltage in TaO$_X$ during forming or set operation, leading to thinner filaments, stronger filament ruptures, and higher on/off ratios[28,29]. The $I_{FORM\_mean}$ reduction (from 2.2 to 1.6 mA) and $R_{LRS}$ increase (from 151 to 185 Ω) of S5 species compared to ones of S2 species (Supplementary Fig. 3d) provide evidence for the formation of thinner filaments.

As a result, by optimizing the oxygen reservoir layer thickness of TaO$_X$ and the O$_2$/Ar gas mixture ratio, the high on/off ratio of 35.2 with slight current reduction (i.e., $R_{LRS}$/$R_{HRS}$ increase to 224/3364 Ω) of CMOS process-compatible TaO$_X$/Ta$_2$O$_5$ memristors was achieved for S5 species. While previous reports achieved current reduction in CMOS-incompatible memristors by adjusting only the O$_2$/Ar ratio[28,30] or only the oxygen reservoir thickness[31,32], our study is, to the best of our knowledge, the first to demonstrate both an enhanced on/off ratio

and a reduced current in CMOS-compatible memristors by simultaneously tuning the O$_2$/Ar ratio and the oxygen reservoir thickness. Meanwhile, the significant reduction in the on/off ratio value of S6 compared to S5 (from 35.2 to 15.6) was due to a dramatic decrease in $R_{HRS}$ value from 6525 to 3364 Ω with a slight increase in $R_{LRS}$ value from 185 to 224 Ω (Fig. 3b, c). In this study, TaO$_X$/Ta$_2$O$_5$ memristors based on 10 nm-thick and 5 nm-thick TaO$_X$ layers with various O$_2$/Ar gas mixture ratios of 3–9% were also fabricated, which showed worse on/off ratio values of 6.4–12.8 and 1–2, respectively, stemming from $R_{HRS}$ degradation. Therefore, it is strongly believed that the on/off ratio reduction of S6 compared to S5 is due to the insufficient oxygen content in TaO$_X$ to function as an oxygen reservoir.

Pulse measurements were performed for the S5 species, which showed the highest on/off ratio value (Fig. 4a). On a long-pulse program/erase (P/E) cycling measurement with a unit pulse time ($T_{PULSE}$) of 50 μs, an on/off ratio value remained above 30 until the number of pulses reached 5000 (Fig. 4b), which satisfied the device requirements for the proposed biosensor transducer (i.e., the on/off ratio and pulse duration of more than 30 and 10 μs, respectively). For reference, a short-pulse P/E cycling measurement with a $T_{PULSE}$ of 2 μs was conducted to check the usability of neuromorphic application, exhibiting a measured on/off ratio value of more than 35 up to several pulses of 30000 (Fig. 4c), which was a good value among the previously reported same CMOS process-compatible memristor stacks of TiN/TaO$_X$/Ta$_2$O$_5$/TiN that did not use noble metals, such as Pt, Au as electrodes[23,25,26]. Meanwhile, another representative CMOS-compatible memristor stack, TiN/Ti/HfO$_2$/TiN[22,33,34], labeled S7, was also fabricated (Fig. 4d) and its long-pulse P/E cycling measurement was performed. It should be noted that the S7 was SET stuck after several pulses of more than 4000 (Fig. 4e), which is attributed to the worse TDDB characteristics of HfO$_2$ compared to Ta$_2$O$_5$. For example, a TDDB value of the HfO$_2$ layer at an electric field (E-field) of $5 \times 10^8$ V/m was measured to be about 33 times shorter than that of the Ta$_2$O$_5$ layer at the same E-field (Fig. 4f).

**Implementation of the memristor-based electrochemical biosensor transducer and its operation verification in a pH sensor**
The memristor-based transducer (Fig. 1a) was practically implemented, as illustrated by the dotted red line in Fig. 5a. Three switches, S/W$_1$, S/W$_2$, and S/W$_3$, were introduced to enable different circuit wiring in the three operational stages: initialization, sensing, and

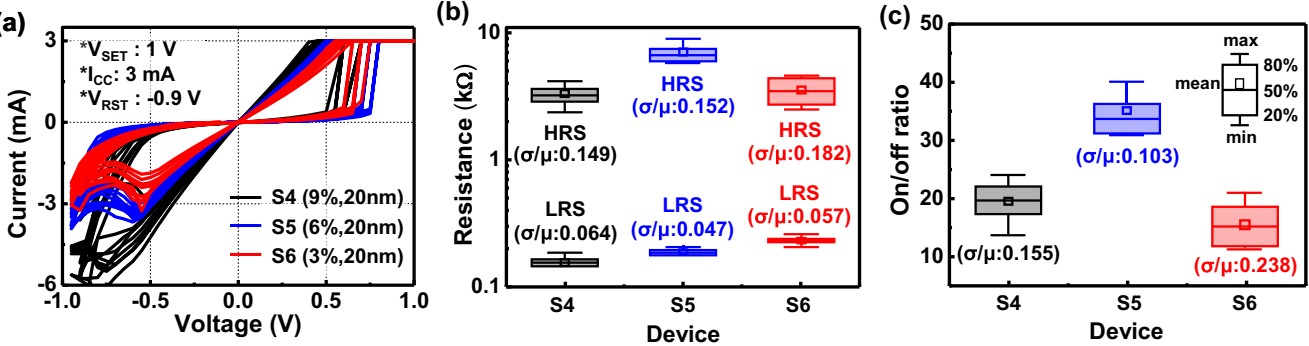

**Fig. 3 | Measured electrical characteristics of CMOS process-compatible TaO$_X$/ Ta$_2$O$_5$ memristors with different TaO$_X$ thicknesses. a–c** Measured I–V curves, LRS/HRS resistances, and on/off ratios of the memristors based on 20 nm-thick TaO$_X$ layers with O$_2$/Ar gas mixture ratios of 9 (S4), 6 (S5), and 3% (S6). The thickness of Ta$_2$O$_5$ layers in S4–S6 species is the same as 4 nm. Ten devices per species were randomly selected on an 8-inch wafer. LRS/HRS resistances were measured at a read voltage of 0.3 V.

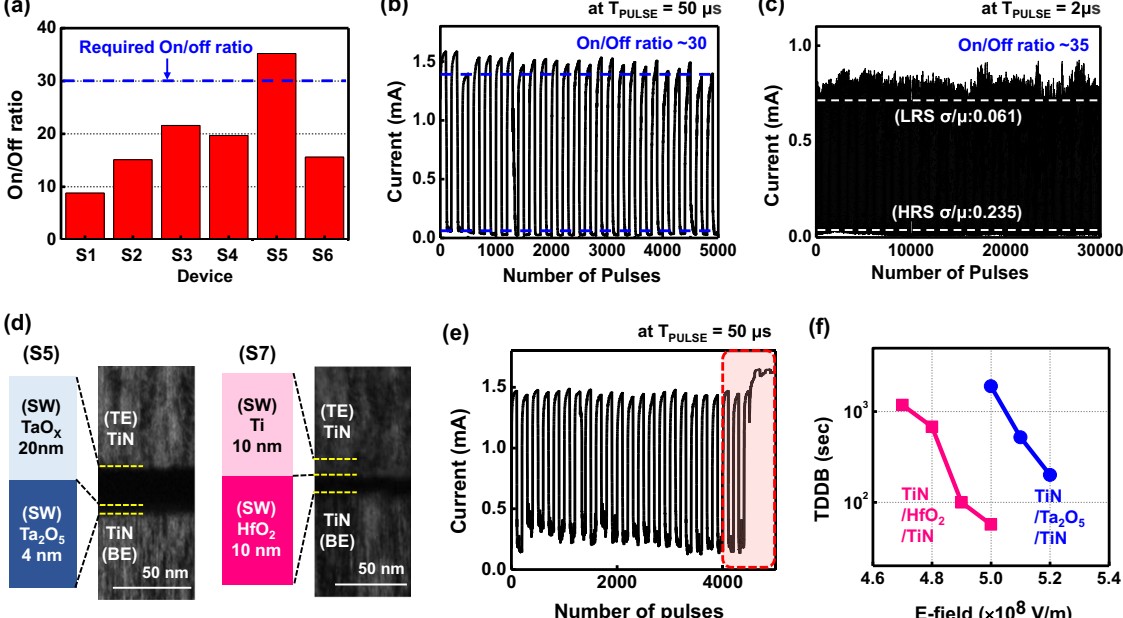

**Fig. 4 | Pulse measurement results of the CMOS process-compatible memristors. a** On/off ratio performance of fabricated all CMOS process-compatible TaO$_X$/ Ta$_2$O$_5$ memristors, indicating that the on/off ratio of S5 species reaches its target value required for the biosensor **b** Measured long-pulse program/erase(P/E) cyclings of the TaO$_X$/Ta$_2$O$_5$ memristor (S5) based on a 20 nm-thick TaO$_X$ layer with an O$_2$/Ar gas mixture ratio of 6% under a condition of a unit pulse time ($T_{PULSE}$) of 50 µs, exhibiting an on/off ratio over 30 until the pulse numbers reaches 4000. One hundred consecutive SET pulses (1 V for 50 µs) followed by 100 consecutive RESET pulses (−1.5 V for 50 µs) are applied to the S5 memristor, and the current value is read by read pulses (0.3 V for 10 µs). **c** Measured short-pulse P/E cyclings of the TaO$_X$/Ta$_2$O$_5$ memristor (S5) under a condition of $T_{PULSE}$ of 2 µs, showing an on/off ratio of more than 35 until the number of pulses of 30000. One hundred consecutive SET pulses (1 V for 2 µs) followed by 100 consecutive RESET pulses (−1.5 V for 2 µs) for a P/E cycling are applied to the S5 memristor, and the current value was read by read pulses (0.3 V for 10 µs). **d** Measured TEM images of two types of fabricated memristors with different switching layer stacks of TaO$_X$/Ta$_2$O$_5$ and Ti/ HfO$_2$. In addition to the TaO$_X$/Ta$_2$O$_5$ memristors of S1–S6, Ti/HfO$_2$ memristors of S7 are also fabricated. The Ti/HfO$_2$ consists of dc sputtered 10 nm-thick Ti and ALD-deposited 10 nm-thick HfO$_2$. **e** Measured long-pulse LTP-LTD curves of the S7 memristor with a $T_{PULSE}$ of 50 µs. The specific pulse conditions are the same as those for S5 in Fig. 4d. **f** Measured time-dependent dielectric breakdown (TDDB) characteristics of TiN/HfO$_2$/TiN and TiN/Ta$_2$O$_5$/TiN devices.

reading. After dropping the analyte solution on the sensor electrode, it took at least 0.5 to 1 min for the electrochemical reaction to stabilize. Repeated accumulation of this reaction time caused deterioration of the device due to voltage and current stresses on the memristors. Therefore, S/W$_1$ was introduced to ensure that these stresses were applied to the memristors only during the sensing stage, which lasts for only a few milliseconds. A parallel two-memristor cell, in which two memristors are connected in parallel, was inserted into the transducer. The variation factor ($σ/µ$) of resistance of the S5 devices reached approximately 20%, as shown in Figs. 3b; 4c and Supplementary Fig. S5. The parallel memristor cell, previously adopted in memory and logic applications[35], was employed to minimize the influence of resistance variation of the transducer, thereby reducing the variation of $V_{OUTR}$, which means a more stable transducer operation (see Supplementary Fig. S6). An electrochemical biosensor system shown in Fig. 5a was designed by integrating a SP block, such as a 12-bit ADC, an MCU, and an organic light-emitting diode (OLED) display with the proposed transducer. ADC monitored the change in $V_{OUT}$, which was processed by the MCU and displayed on the OLED. The MCU not only determined the R$_M$ value by processing the $V_{OUT}$ but also controlled the transducer. The timing diagram and detailed operation of the biosensor are depicted and explained in Fig. 5b, respectively.

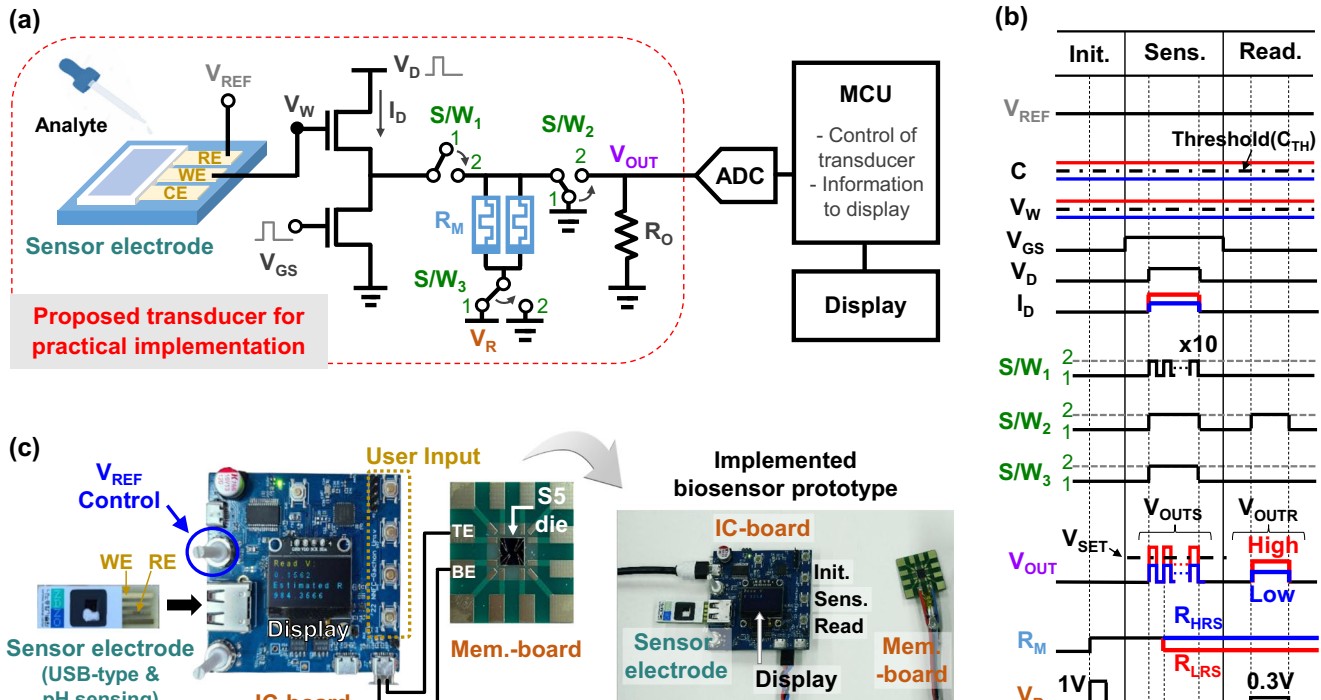

**Fig. 5 | Block, timing diagrams, and implementation of a proposed electro-chemical biosensor system. a** A block diagram of the system, consisting of the proposed memristor-based transducer for practical implementation, a signal pro-cessing (SP) block, such as an ADC, an MCU, and a display. **b** A timing diagram of the biosensor, comprising initialization (init.), sensing (sens.), and reading (read.) stages. In the initialization stage, the switches of S/$W_1$, S/$W_2$, and S/$W_3$ are con-nected in positions 1, 1, and 1, respectively. 1 V was applied to $V_R$ for 40 ms, initi-alizing the memristors to a high resistance state (HRS). During the sensing stage, S/$W_1$, S/$W_2$, and S/$W_3$ were connected in positions 2, 2, and 2, respectively. More precisely, S/$W_1$ was switched to position 2 only during ten consecutive pulses with a unit pulse time ($T_{PULSE}$) of 20 µs to minimize voltage stress on the memristors. 1.2 V was applied to $V_D$, along with the necessary voltages to $V_{REF}$ and $V_{GS}$. If the voltage ($V_{OUT}$) across the memristors exceeded the threshold level of about 0.7 V ($V_{SET}$), the memristor shifts to a low resistance state (LRS); otherwise, it remained in HRS. In the reading stage, S/$W_1$, S/$W_2$, and S/$W_3$ were connected in positions 1, 2, and 1, respectively. 0.3 V was applied to $V_R$ for 80 µs. A voltage drop occurred across the memristors, and the potential at $V_{OUT}$ was sensed through the ADC and processed via the MCU to determine the memristor resistance of $R_M$. **c** Images of an imple-mented prototype of the biosensor system, comprising a sensor electrode, an IC-board, a mem.-board, and an organic light-emitting diode (OLED) display.

A prototype of the fully-integrated biosensor system was imple-mented, consisting of a sensor electrode, an IC-board, a mem.-board, and an OLED display (Fig. 5c). The sensor electrode was fabricated in a disposable USB form and consisted of a reference electrode and a working electrode, on which patterned Ti/Al metals were formed to function as a potentiometric pH sensor. The mem.-board consisted of a fabricated 5 × 5 mm² single die with S5 memristors (named as S5 die), which was mounted on a printed circuit board (PCB). The IC-board consisted of the proposed transducer excluding the sensor electrode and memristors, the SP block, and various components, such as operational amplifiers, LDO regulators, capacitors, resistors, switches, and buttons. Several USB ports were provided for interconnection with the sensor electrode, power supply, and UART communication. Two potentiometers were included for controlling $V_{REF}$. All operations were user-controlled via buttons. When the three user buttons of init., sens., and read. were pressed, the display provided information about the device's $R_M$ and the IC's $V_{OUT}$ values. The implementation of the bio-sensor system is described in detail in the "Methods" section.

The functionality of the transducer itself was evaluated by applying an external voltage to the $V_W$ node, i.e., instead of con-necting the sensor electrode to the transducer. With $V_W$ and $V_{GS}$ each fixed to 2 V, the $V_{OUT}$ in the sensing stage ($V_{OUTS}$) was measured by applying a pulse train to position 2 of S/$W_1$, which consisted of ten consecutive pulses with a $T_{PULSE}$ of 20 µs, exhibiting that the $V_{OUTS}$ slightly decreased from 0.69 to 0.62 V up to the eighth pulse (Fig. 6a). This decrease in $V_{OUTS}$ indicated successful resistance ($R_{MP}$) transition from HRS to LRS of the parallel memristor cells. The $V_{OUTS}$ was given by $I_D \times R_{EQ}$, where $R_{EQ}$ is $R_T//R_{MP}//R_O$, as shown in Fig. 6b.

Although $R_T$ was designed to be small, less than 20 Ω, $R_{EQ}$ decreased by 10 to 20% with the change in $R_{MP}$ from HRS to LRS, resulting in a slight decrease in $V_{OUTS}$. As $V_W$ was swept from 1.4 to 2.2 V, the changes in $V_{OUTS}$ and $R_{MP}$ were observed (Fig. 6c). $V_{OUTS}$ transitioned linearly at $V_W$ values from 1.4 to 2 V, following the transfer curve characteristic of the upper FET designed to operate in the linear region (Fig. 6b). When $V_W$ changed from 2.0 to 2.1 V, $R_{MP}$ decreased abruptly from an $R_{HRS}$ of about 700 Ω to an $R_{LRS}$ of ~80 Ω, along with a decrease in $V_{OUTS}$ from 0.75 to 0.62 V. When $V_W$ changed from 2.1 to 2.2 V, $R_{MP}$ maintained the $R_{LRS}$. This abrupt $R_{MP}$ transition, with a clear transition point at $V_W$ of 2.1 V, was an experimental result that demonstrated the TS function of the implemented transducer. Although the memristors exhibited device-to-device resistance var-iation, the resistance parallelization scheme, by employing variation-tolerant $R_{EQ}$ instead of directly using $R_{MP}$, effectively suppressed this variation and enabled a stable $V_{OUTS}$ response with a clearly defined resistance switching characteristic of the parallel memristor cell (see Supplementary Figs. S5 and S7).

The unique operation of the proposed transducer-based bio-sensor system was verified in a pH sensor platform by connecting the sensor electrode to the system, i.e., the WE to the $V_W$ node and the RE to the $V_{REF}$ node. With the $V_{REF}$ fixed at a value, i.e., 1.9 V, perfor-mance parameters of the system, such as $V_W$, $V_{OUTS}$, $R_M$, and $V_{OUT}$ at the reading stage ($V_{OUTR}$), were measured in response to the change in pH value (Fig. 6d; Supplementary Fig. S8 and Supplementary Video 1). As the pH decreased from 10 to 2 (i.e., as the acidity increased), the $V_W$ increased linearly from 1.76 to 2.22 V, exhibiting a close-to-linear behavior with a slope of −57 mV/pH, which is within

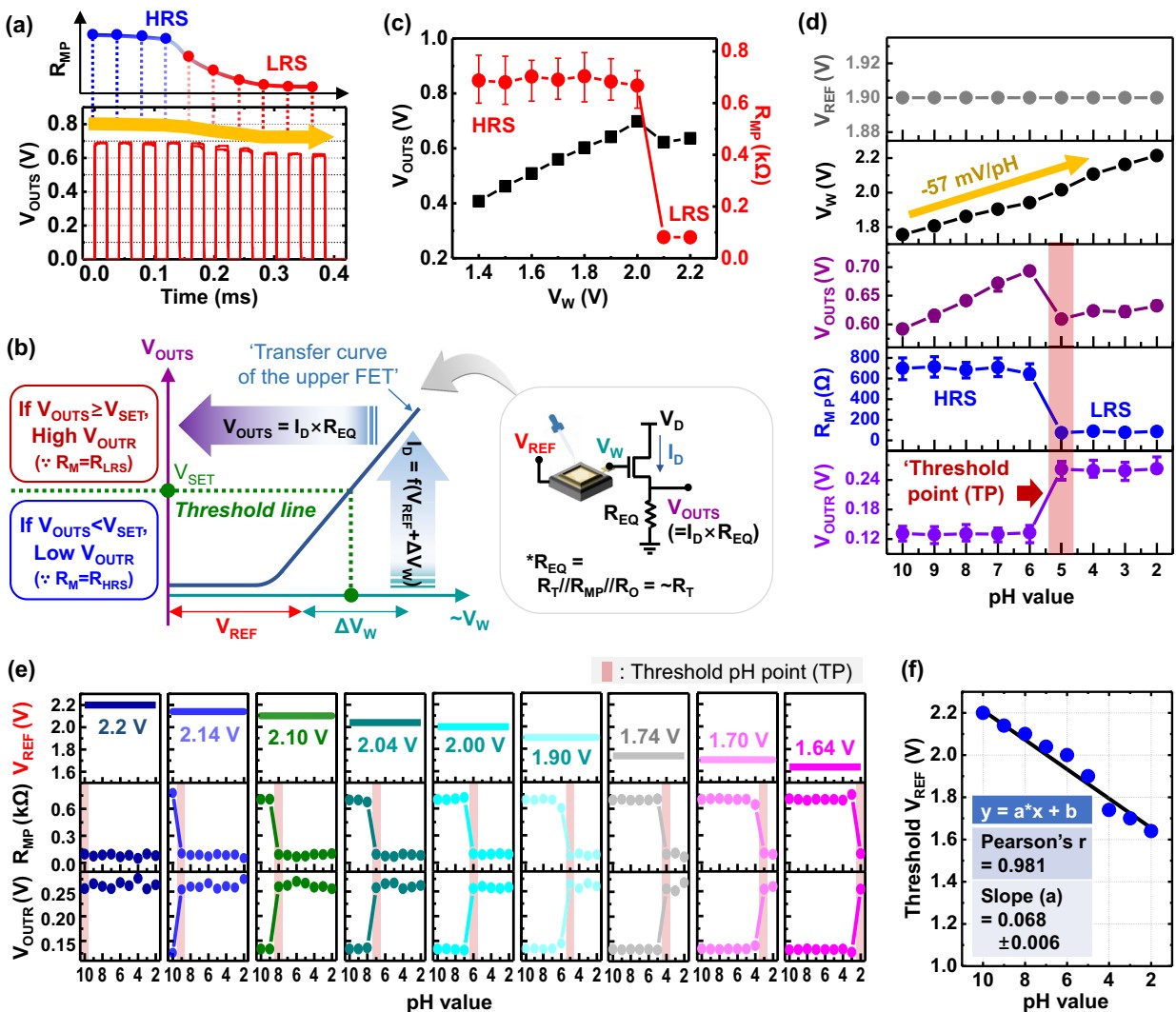

**Fig. 6 | Experimental demonstration of the proposed electrochemical biosensor system. a** Measured output voltage waveform at the sensing stage ($V_{OUTS}$) of the system without the sensor electrode, when the working electrode voltage ($V_W$) and a $V_{GS}$ were both fixed at 2 V. The $V_{OUTS}$ value decreased up to the eighth pulse of a total of ten pulses, indicating a noticeable change in the parallel memristor resistance ($R_{MP}$) to LRS. **b** Theoretical operation mechanism of the proposed transducer. $\Delta V_W$ is the potential difference in $V_W$ before and after dropping pH solutions (pH 2 and 10) onto a fabricated USB-type sensor electrode, $I_D$ is the drain current of the upper FET, and $V_{OUTR}$ is the output voltage waveform at the reading stage. **c** Measured $V_{OUTS}$ and $R_{MP}$ characteristics with respect to $V_W$ of the system without the sensor electrode, indicating that the memristors remain in HRS at a low $V_W$, and the transition to LRS occurs at 2.0 V. **d** pH measurement results of the system under a fixed $V_{REF}$ of 1.9 V, indicating that the threshold pH point (TP) of the pH sensor is pH 5. **e** pH measurement results of the system with respect to $V_{REF}$, showing that the TP shifts in units of pH 1 by controlling the $V_{REF}$ value. **f** Threshold $V_{REF}$ versus pH characteristic of the system, exhibiting good linearity with a Pearson's correlation coefficient of 0.985. The threshold $V_{REF}$ is defined as the $V_{REF}$ corresponding to each TP. Experimental repetitions: **a**, **c**, **d** were obtained from different memristor cells ($n = 3$), and **e**, **f** from a memristor cell ($n = 1$). Error bars indicate min–max values ($n = 3$).

the range of the slope values of previously reported pH sensors[36–39] and is close to the theoretical Nernst value for the extended-gate FET pH sensor (−59.12 to −59.16 mV/pH)[38,39]. $R_{MP}$ and $V_{OUTR}$ were maintained at HRS (above 600 Ω) and low voltage (<140 mV) for pH values from 10 to 6, and at LRS (below 90 Ω) and high voltage (above 260 mV) for pH values from 5 to 2. When the pH decreased from 6 to 5, $R_{MP}$ transitioned from HRS to LRS, and the corresponding $V_{OUTR}$ changed from low to high voltage, experimentally indicating a threshold pH point (TP) for pH sensing was pH 5.

Because $V_W$ equals the sum of $V_{REF}$ and potential difference in $V_W$ ($\Delta V_W$), which are input parameters of the system, is $V_W$, where $V_W$ is linearly proportional to $V_{OUTS}$ (Fig. 6b), $V_{REF}$ can serve as an input variable that controls TP. Figure 6e shows pH measurement results of the system with respect to $V_{REF}$, exhibiting that the TP changed from

pH 10 to 2 in step of 1 pH 1 unit increments when $V_{REF}$ was changed from 2.2 to 1.64 V. A TP versus corresponding $V_{REF}$ (called threshold $V_{REF}$) graph showed good linearity with a Pearson's correlation coefficient of 0.981 (Fig. 6f). The slope deviation of about 6 mV/pH in the graph was attributed to the variation in the sensor electrode utilized. The graph strongly correlated with another graph for $\Delta V_W$ before and after dropping pH solutions with values between 2 and 10 onto a fabricated sensor electrode (Supplementary Fig. S9). In this study, no surface treatment was performed on the Ti/Au electrodes, and thereby the slope deviation could be significantly reduced through additional surface treatment of the electrode[40,41]. Furthermore, to substantiate the applicability of the proposed memristor technology towards electrochemical biosensing, we extended its use to the detection of glucose and ascorbic acid with appropriate sample preprocessing (see Supplementary Figs. S10–S12; Supplementary Videos 2 and 3). The

memristor device successfully switched its resistance state upon reaching the predefined threshold concentrations of these analytes, further validating the demonstration of the proposed memristor-based biosensor with real biomarkers.

In addition, we implemented a simplified version of the biosensor by replacing the bulky components of the ADC and display with a comparator, hold circuit, and LED (Supplementary Fig. S13a). The comparator detected and compared the $V_{OUTR}$ to a 0.2 V reference voltage during the read stage. When the user pressed the button for a read operation, if $V_{OUTR}$ was less than 0.2 V, the memristor went into the HRS; otherwise, $V_{OUTR}$ went high, and a hold circuit maintained this voltage for approximately 1 s, causing the LED to turn on (Supplementary Fig. S13b). The user could easily check the status of the memristor repeatedly by pressing the button for the read stage. The TP value of the pH sensor was successfully detected at intervals of 1 pH unit by checking the LED turning on and off while varying the change in $V_{REF}$ (Supplementary Fig. S13c). The simplified system achieved approximately a 4-fold reduction in IC board size and a 12.4-fold reduction in power consumption, relative to the fully-integrated version (Supplementary Table S1).

Our sensor technology is compared with previously reported semiconductor-based sensors (see Supplementary Table S2). To the best of our knowledge, the sensors developed in this work are the first demonstration of a semiconductor device-based electrochemical biosensor that exhibits TS functionality at the transducer level together with complete system-level integration. Unlike prior transducing devices, our memristor device also offers full CMOS compatibility, which is readily extendable to monolithic integration with SP blocks and thereby facilitating the realization of miniaturized sensor systems. Moreover, our transducer is the only one that simultaneously exhibits intrinsic TS functionality and long-term non-volatility (Supplementary Fig. S14), which is potentially valuable for miniaturized sensors with self-diagnosis capability requiring data retention.

Owing to its simplicity, scalability, and functionality, the proposed memristor-based technology shows promise for future integration into portable sensing platforms, including point-of-care and health-monitoring systems. Furthermore, in miniaturized electronic devices, such as ingestible capsule prototypes, this approach could offer advantages, such as miniaturization, non-volatile memory, scalability, and ultra-low power consumption, potentially contributing to future efforts aimed at monitoring internal physiological environments in a safe and minimally invasive manner (Supplementary Fig. S15).

Although this study was conducted as a proof-of-concept for the proposed biosensor transducer, effective control of $V_{SET}$ variation of the memristors is critical for ensuring reliable biosensor operation. Possible mitigation strategies at the device, circuit, and system levels are summarized in Supplementary Fig. S16 and Supplementary Note 1.

Furthermore, further efforts are required to validate the sensing functionality under realistic biochemical conditions. The proposed memristor-based transducer was evaluated for biosensing applications using pH, glucose and ascorbic acid, as representative analytes. Among these, glucose detection currently operates effectively only under strong alkaline conditions, indicating the need for further material and interface optimization. Future studies should therefore assess device performance in complex biofluid environments to understand the influence of factors such as biofouling, signal drift, and cross-sensitivity. Ultimately, comprehensive validation with clinically relevant samples under real-world operating conditions will be essential to establish the broader translational potential of this technology.

## Discussion

In this study, we demonstrated a bio-to-electrical transducer with TS functionality by leveraging a memristor with single-bit memory

characteristics for electrochemical biosensor applications. To this end, a CMOS-compatible RRAM-based $TaO_X/Ta_2O_5$ memristor was developed. By tuning the oxygen reservoir layer in $TaO_X$ with an $O_2/Ar$ gas mixture ratio of 6% and a thickness of 20 nm, the memristor achieved a high on/off ratio (>30) and maintained a long unit pulse width exceeding 10 μs, meeting the device requirements for the intended operation of the proposed transducer. Using this memristor, a complete electrochemical biosensor system was implemented by integrating a sensor electrode, an SP module, and a display with the proposed transducer. The intended TS function of the proposed transducer was experimentally demonstrated. With the $V_{REF}$ fixed to a value, i.e., 1.9 V, $R_M$ switches from HRS (>600 Ω) to LRS (<90 Ω) at a TP of pH 5, with the corresponding output voltage shifting immediately from low (<140 mV) to high (>260 mV) values. These experimental results confirm the feasibility of the proposed memristor devices for electrochemical biosensing applications. Furthermore, by varying $V_{REF}$ from 1.64 to 2.2 V, the TP was tunable to any desired pH value between pH 2 and 10 in step of 1, showing strong linearity between TP and pH with a Pearson's correlation coefficient of 0.981. In addition, the biosensor architecture was streamlined by substituting the bulky ADC and display modules with a comparator, a hold circuit, and an LED, enabling a compact and portable device design. To the best of our knowledge, this memristor-based system represents the first experimental demonstration of a semiconductor device-based electrochemical biosensor that integrates TS functionality directly at the transducer level while preserving system-level complexity. Owing to its simplicity, scalability, and functionality, the proposed memristor-based technology shows promise for future integration into portable sensing platforms and miniaturized devices that could benefit from inherent memory retention.

## Methods

### Device fabrication

A 100 nm $SiO_2$ layer on 8-inch Si wafers was thermally oxidized in a furnace. A BE stack of Ti/Al/Ti/TiN with a thickness of 10/450/10/120 nm was sputtered on the $SiO_2$ layer. A 4 nm-thick $Ta_2O_5$ layer was deposited on the sputtered TiN BE by thermal ALD at 200 °C using a tantalum precursor, TBTDET. The $TaO_X$ layers with varying thicknesses of 5, 10, 20, and 40 nm and $O_2/Ar$ gas mixture ratios of 3, 6, and 9% were deposited on the $Ta_2O_5$ layer using a reactive sputtering system at 400 °C. A 12 nm TiN TE layer was sputtered on the $TaO_X$ layer. The active region (4 μm²) was defined by dry etching with $Cl_2$ and $BCl_3$ gases. Subsequent processes followed the standard back-end-of-line procedure, as shown in Fig. S1c. An 800 nm-thick inter-metal dielectric layer of $SiO_2$ was formed by high-density plasma chemical vapor deposition with a process temperature at 350 °C, plasma-enhanced CVD at 400 °C, and chemical mechanical polishing. Metal vias and plugs were sequentially formed by dry etching with $CF_X$ gases and tungsten-based molecular organic CVD at 300 °C. Finally, the metal interconnection (M1) of the Ti/Al/Ti/TiN stack was patterned to form TE and BE pads for contacting microprobes during electrical measurements.

### Biosensor implementation

The memristor-based biosensor system consists of a USB-type sensor electrode, a mem.-board, an IC-board, and a display as shown in Fig. 5c. For the USB-type sensor electrode, an 800 nm-thick $SiO_2$ layer was thermally oxidized on an 8-inch Si substrate, 20 nm-thick Ti and 200 nm-thick Au layers were patterned using an evaporator, optical lithography, and dry etching processes. The wafer with the patterned Ti/Au metals was then diced into 5 × 5 mm² chip including electrodes. Each chip was installed on a PCB substrate using wire bonding and epoxy passivation in a USB-A form factor. For the mem.-board, an 8-inch wafer with S5 memristors was diced into 5 × 5 mm² (named as S5

die), and Au electrodes were formed on a 30 × 30 mm² PCB. TE and BE electrodes of the S5 die and the Au electrodes of the PCB were connected through Au wire bonding. Two memristors used in the parallel two-memristor cell configuration were interconnected on the S5 die through a standard metal formation process involving sequential photolithography and metal deposition steps, in which two TEs and two BEs were connected to each other. Electrical components on the IC board include nMOS transistors (ALD212902SAL, Advanced Linear Devices) for forming $I_D$ and $R_T$ of the transducer and an MCU (nRF52832, Nordic Semiconductor) for SP, operational amplifiers (LMP7721MA, Texas Instruments), linear voltage regulators (LTC3025, Analog Devices), diodes, resistors, capacitors, buttons, switches, and various connectors. The IC board was assembled on a 2-layer FR4 PCB, with each component placed using a soldering iron (FX-951, Hakko) and a heat gun (861DW, Quick). To implement a prototype of the biosensor, the mem.-board and the IC-board were connected with alligator clips (JM-013), a commercial OLED display (SSD1306, Solomon Systech), and a 5 V charger were used. The firmware was developed in Segger Embedded Studio and uploaded to the MCU for portability

## Data availability

The data generated in this study are provided in the Source data file and openly available in figshare at https://doi.org/10.6084/m9.figshare.30126289. Source data are provided with this paper.

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

## Acknowledgements

This work was supported by the Nanomedical Devices Development Project of NNFC in 2024, supported by Next-generation Intelligence semiconductor R&D Program through the National Research Foundation of Korea (NRF) funded by the Korea government (MSIT) (RS-2024-00406897), supported by National R&D Program through the National Research Foundation of Korea (NRF) funded by Ministry of Science and ICT (RS-2025-24683075), and supported by the Korea Evaluation Institute of Industrial Technology (KEIT) funded by the Ministry of Trade, industry and Energy (MOTIE) (RS-2024-00508418 and RS-2024-00438660).

## Author contributions

Y.J.K., Y.K., and Y.Y. contributed equally to this work. Y.J.K. and J.L. conceived and designed the research. Y.J.K., Y.K., Y.Y., K.T., S.S., A.J., N.H.B., W.C.L., G.S., H.C., M.H.K., Y.K., and J.L. developed the methodology. Y.J.K., Y.K., Y.Y., A.J., and J.L. curated the data and visualized the results. Y.L., S.P., Y.P., K.G.L, C.Y.K., H.K., W.S.S, S.J.L., J.H.A., and S.C. provided critical insights. Y.J.K., Y.K., and J.L. wrote the original draft. Y.J.K., S.C., K.M.K., and J.L. supervised the work.

## Competing interests

The authors declare no competing interests.

## Additional information

[1]Department of Electronic Engineering, Gachon University, Seongnam, Republic of Korea. [2]Department of Semiconductor Engineering, Gachon University, Seongnam, Republic of Korea. [3]Division of Nano Convergence Technology Development, National Nanofab Center (NNFC), Daejeon, Republic of Korea. [4]Department of Materials Science and Enginnering,, Chungnam National University (CNU), Daejeon, Republic of Korea. [5]Department of Electronics Engineering, Chungnam National University (CNU), Daejeon, Republic of Korea. [6]Department of Materials Science and Engineering, Korea Advanced Institute of Science and Technology (KAIST), Daejeon, Republic of Korea. [7]School of Electrical Engineering, Korea Advanced Institute of Science and Technology (KAIST), Daejeon, Republic of Korea. [8]Department of Semiconductor Convergence, Chungnam National University (CNU), Daejeon, Republic of Korea. [9]These authors contributed equally: Young-Joon Kim, Youna Kwon, Youngwoo Yoo. ✉e-mail: youngkim@gachon.ac.kr; shinhyun@kaist.ac.kr; km.kim@kaist.ac.kr; jwlee80@cnu.ac.kr

