## [Transparent Peer Review File · Nature Communications]

Demonstration of CMOS-compatible Memristor-based Electrochemical Biosensor Transducer with Threshold-sensing Functionality

Corresponding Author: Professor Jongwon Lee

Version 0:

Reviewer comments:

Reviewer #1

(Remarks to the Author)

This paper proposes a novel bioelectrical signal transducer with threshold-sensing (TS) functionality, enabling electrochemical biosensing applications by utilizing a memristor with single-bit memory characteristics. To meet device requirements for the transducer (achieving a high on/off ratio exceeding 30 while maintaining a long pulse width exceeding 10 milliseconds), a CMOS-compatible TaOX/Ta₂O₅ memristor was developed through process optimization – specifically, setting the oxygen-storage TaOX layer with an O₂/Ar gas mixture ratio of 6% and a thickness of 20 nm. Subsequently, a complete electrochemical biosensor system was constructed by integrating sensing electrodes, signal processing modules, and displays based on the optimized memristor. Experimental verification confirmed the TS functionality: by adjusting VREF from 1.64 V to 2.2 V, the threshold pH point (TP) could be precisely tuned across the pH 1-10 range in 1-pH increments, demonstrating excellent linearity between TP and pH values (Pearson correlation coefficient = 0.981). Furthermore, a more streamlined biosensor version was implemented by replacing bulky ADC and display components with a comparator, hold circuit, and LED. The proposed memristor-based system technology holds some application potential in portable point-of-care diagnostics. Before recommend it for publication, there some questions that need to be addressed.

Q1: Should the reference to 'Fig. S1' in line 201 on page 10 be corrected to 'Fig. S2'? This is because the forming voltage and forming current mentioned in line 200 are actually presented in Fig. S2.

Q2: Regarding the caption of Fig. 5 on page 14, which mentions the use of 'only ten consecutive pulses with a unit pulse time (TPULSE) of 20 μs to minimize voltage stress to the memristors' during the sensing stage – how were the three parameters (20 μs pulse width, 10 pulses, and 80 μs duration in the reading stage) experimentally determined? Why is 10 pulses considered the optimal solution for stress minimization?

Q3: In the schematic diagram in the first half of Fig. 5(c) on page 14, the IC board is shown connected to the four terminals (BE1, TE1, BE2, TE2) of the memristor chip via serial ports. However, in the photograph in the latter half, only two alligator clips are used to establish the connection. What explains this discrepancy in connection methods?

Q4: Regarding lines 264-265 on page 15 stating 'To minimize the influence of the resistance variation on the transducer, the parallel memristor cell was chosen, which has been utilized in the memory and logic fields' - what specific advantages does this dual-memristor configuration offer compared to the previously mentioned single-memristor structure? How does this architectural improvement enhance the performance of your system?

Q5: Line 387 on page 18 mentions that 'the simplified biosensor version using a comparator, hold circuit, and LED to replace bulky ADC and display components could still accurately detect threshold points at pH 1 intervals.' However, the more complex version with ADC and MCU also achieved threshold detection at pH 1 intervals. Could the complex version potentially achieve higher detection precision? What are the advantages of the simplified version over the complex one? What is the performance gap between these two versions? We recommend adding comparative experiments.

Q6: Could you please supplement an experiment to demonstrate the real-time detection capability of the complex version (with ADC and MCU)?

Q7: Is the memristor strictly necessary for this work? Could other devices potentially replace the memristor's functionality, given that memristor chip fabrication is relatively complex?

Reviewer #2

(Remarks to the Author)

In this paper, authors report the CMOS-compatible memristor-based electrochemical biosensor transducer with threshold-sensing functionality. The revision suggestions are as follows:

- (1) The application of sensing effect based on memristor has been reported in previous studies, and the author should highlight the innovation of this work by comparing it with the results reported previously.
- (2) The production and quality of figures need to be further improved.
- (3) The formation of conductive filaments requires relevant experimental evidence, such as in situ TEM images.
- (4) What are the important and unique applications of this work compared to other related reports (such as: Nature Communications, 2023, 14, 8489)? The author should clarify in the main text.
- (5) English grammar and writing errors require careful correction by the author.

Reviewer #3

(Remarks to the Author)

This paper reports on a TaO_x/Ta₂O₅ memristor device designed for biomarker threshold-sensing applications. The authors conducted extensive research on memristor performance and demonstrated the concept by integrating a sensor electrode for pH sensing, a signal processing (SP) module, and a display with the proposed transducer. However, the pH demonstration falls short of illustrating the biomarker detection capabilities mentioned in the introduction, such as C-reactive protein detection, which presents significantly greater technical challenges for accurate measurement. The authors' contributions primarily focus on memristor characterization. Given that TaO_x/Ta₂O₅ materials have been widely applied in memristors and numerous reports on biomarker threshold detection already exist (Nature Electronics, 2023, 6, 765–770; Nature Electronics, 2024, 7, 1176–1185; Nature Communications, 2025, 16, 4334, among others), the novelty presented in this paper is insufficient for publication in Nature Communications. The following suggestions are provided to improve the manuscript:

1. It is suggested that the authors propose a specific application scenario that is well-suited to pH threshold-biosensing capabilities.
2. Using only pH sensing as proof-of-concept fails to demonstrate relevance to actual diagnostic applications. Real biomarkers (glucose, troponin, etc.) in complex biological matrices are essential for meaningful validation.
3. Device variability maybe the critical obstacle to practical applications. The stochastic nature of RRAM could lead to false positive result. Device-to-device variability characterization is insufficiently addressed in your work. Additionally, what quality control measures do you implement to ensure acceptable device-to-device consistency for reliable biosensor operation?
4. Clinical samples contain numerous potential interferents that could compromise specificity. Interference testing is needed.
5. Error bars are missing in Fig 6. Were these measurements obtained from a single device, or do they represent averaged data from multiple independent devices?
6. Will temperature and humidity affect the stability of the sensor? What environmental compensation strategies will you implement?

Reviewer #4

(Remarks to the Author)

Version 1:

Reviewer comments:

Reviewer #1

(Remarks to the Author)

The authors have answered all my comments. I can recommend it for publication in Nature Communications.

Reviewer #2

(Remarks to the Author)

This revised manuscript can be accepted for publication.

Reviewer #3

(Remarks to the Author)

The authors have addressed most of my comments. The revised manuscript is much clearer in its novelty. Just a minor point: Although this work was conducted as a proof-of-concept study for the proposed biosensor transducer, detecting glucose and AA in a strong alkaline environment is not suitable for physiological conditions.

Reviewer #4

(Remarks to the Author)

Response Letter to Reviewers' Comments

We sincerely appreciate the reviewers' time and effort in evaluating our manuscript and for providing constructive comments and suggestions that have helped improve the quality of our work. In response to the reviewers' evaluations, we have made a point-by-point response to the reviewers' comments and included additional experimental and simulation results. We have also revised the manuscript to improve the clarity of the manuscript to present our findings more effectively. Based on the responses below, we have updated 2 figures in the revised manuscript, added 10 supplementary figures, 2 supplementary tables, 1 supplementary note, and 3 supplementary videos in the revised Supplementary Information. All modifications and additions in the revised manuscript and Supplementary Information are highlighted in red. Our detailed point-by-point responses to the reviewers' comments are as follows.

Reviewer #1

This paper proposes a novel bioelectrical signal transducer with threshold-sensing (TS) functionality, enabling electrochemical biosensing applications by utilizing a memristor with single-bit memory characteristics. To meet device requirements for the transducer (achieving a high on/off ratio exceeding 30 while maintaining a long pulse width exceeding 10 milliseconds), a CMOS-compatible TaO_x/Ta₂O₅ memristor was developed through process optimization – specifically, setting the oxygen-storage TaO_x layer with an O₂/Ar gas mixture ratio of 6% and a thickness of 20 nm. Subsequently, a complete electrochemical biosensor system was constructed by integrating sensing electrodes, signal processing modules, and displays based on the optimized memristor. Experimental verification confirmed the TS functionality: by adjusting V_{REF} from 1.64 V to 2.2 V, the threshold pH point (TP) could be precisely tuned across the pH 1-10 range in 1-pH increments, demonstrating excellent linearity between TP and pH values (Pearson correlation coefficient = 0.981). Furthermore, a more streamlined biosensor version was implemented by replacing bulky ADC and display components with a comparator, hold circuit, and LED. The proposed memristor-based system technology holds some application potential in portable point-of-care diagnostics. Before recommend it for publication, there some questions that need to be addressed.

Response: We sincerely appreciate the reviewer's constructive comments on our work and valuable suggestions for enhancing the quality of the study.

Comment #1:

Should the reference to 'Fig. S1' in line 201 on page 10 be corrected to 'Fig. S2'? This is because the forming voltage and forming current mentioned in line 200 are actually presented in Fig. S2.

Response: We greatly appreciate the reviewer for pointing out the typographical error. As the reviewer mentioned, the sentence has been corrected in the revised manuscript as follows:

Page10, line 196: The reduction in forming voltage ($V_{\text{FORM_mean}}$) and current ($I_{\text{FORM_mean}}$) of S3 compared to ones of S1 (Supplementary Fig. S2) is a piece of evidence supporting the generation of the thinner filaments.

Comment #2:

Regarding the caption of Fig. 5 on page 14, which mentions the use of 'only ten consecutive pulses with a unit pulse time (T_{PULSE}) of 20 μs to minimize voltage stress to the memristors' during the sensing stage – how were the three parameters (20 μs pulse width, 10 pulses, and 80 μs duration in the reading stage) experimentally determined? Why is 10 pulses considered the optimal solution for stress minimization?

Response: We sincerely thank the reviewer for this valuable comment. As this study focused on experimentally demonstrating the unique threshold-sensing functionality of the biosensor proposed here for the first time, rather than aiming to demonstrate its high performance, the pulse width and number of pulses were not tightly constrained. The detailed explanations are shown below:

1) Rationale for selecting a unit pulse width (T_{PULSE}) of 20 μs in the sensing stage

In this study, we utilized a readily available commercial ADC, specified in its datasheet with a sampling rate of 200 kilo samples per second (200 kS/s). This specification corresponds to a

theoretical pulse width of 5 μs (i.e., 1/200,000 s). In practical ADC design, however, it is generally recommended to employ a pulse width two to three times longer than the theoretical value to ensure stable sampling, which, in the context of this work, corresponds to 10-15 μs . Consequently, a unit pulse width (T_{PULSE}) for the sensing stage was set to 20 μs , encompassing the recommended 10-15 μs range.

2) Rationale for selecting 10 pulses in the sensing stage as the optimum for stress minimization

As shown in Fig. 6(a) of the manuscript, under the condition where SW_1 had a unit pulse width (T_{PULSE}) of 20 μs , the output voltage in the sensing stage (V_{OUTS}) gradually decreased from 0.69 V to 0.62 V during the first eight pulses and remained constant thereafter. Therefore, the minimum number of pulses required for the memristor to transition from the high-resistance state (HRS) to the low-resistance state (LRS) was found to be eight. Although the number of pulses could have been set to eight or nine, it was set to ten to ensure reliable operation even in the unlikely event of an insufficient number of pulses. As the focus of this study was on functional verification, we did not attempt to reduce the number of pulses to eight or nine.

To address the reviewer's comments, a related sentence has been corrected in the revised manuscript as follows:

Page 16, line 282: ~~ slightly decreased from 0.69 to **0.62 V up to the eighth** pulse (Fig. 6a) ~

3) Rationale for selecting an 80 μs pulse width in the reading stage

As this study was focused on experimentally validating the operation of the proposed biosensor rather than achieving high performance, no particular attempt was made to reduce the pulse time. For the endurance evaluation of the memristor used, as shown in Figs. 4b and 4c, the read condition was 0.3 V with a pulse width of 10 μs . Accordingly, the read pulse width for the biosensor in this work was set to a sufficiently larger value of 80 μs .

We found a typo in the manuscript. In the figure caption of Figs. 4b and 4c on page 12, "read pulses (3 V for 10 μs)" should be corrected to "read pulses (0.3 V for 10 μs)." We have revised the manuscript to correct it as follows:

Page 12, lines 6 & 10 in Fig. 4 caption: read pulses (**0.3 V** for 10 μs)

Please let us know if you think that any additional information and measurements related to

the points mentioned above should be provided. We are willing to make the relevant changes.

Comment #3:

In the schematic diagram in the first half of Fig. 5(c) on page 14, the IC board is shown connected to the four terminals (BE1, TE1, BE2, TE2) of the memristor chip via serial ports. However, in the photograph in the latter half, only two alligator clips are used to establish the connection. What explains this discrepancy in connection methods?

Response: We sincerely thank the reviewer for this valuable comment and for providing us with the opportunity to correct the erroneous figure. The two memristors were electrically interconnected directly on the raw wafer (S5 die), not on the memristor board (Mem.-board), via a standard metal formation process comprising sequential steps of photolithography and metal deposition. Their top electrodes were interconnected to each other, and likewise, their bottom electrodes were interconnected to each other, as shown in Figure R1. Accordingly, the wiring between IC-board and Mem.-board shown in Fig.5c was inaccurately depicted in the original manuscript. Figure R1 shows the previous and revised figures shown below.

To address the reviewer's comments, we have updated Fig. 5c with the revised figure in the revised manuscript. Additional explanations have been appended to clarify the metal interconnection of the parallel two-memristor cell, as follows:

Page 21, line 402: **Two memristors used in the parallel two-memristor cell configuration were interconnected on the S5 die through a standard metal formation process involving sequential photolithography and metal deposition steps, in which two TEs and two BEs were connected to each other.**

Previous figure (Figure 5c in the original manuscript)

Revised figure (Figure 5c in the revised manuscript)

Figure R1. Metal interconnection for the parallel two-memristor cell. The top figure presents the circuit diagram and the 3D structure of the parallel two-memristor cell. The red line inserted in the figure indicates the interconnection metal line, representing the parallel connection of the two memristors on the raw wafer (S5 die). Accordingly, the wiring between the Mem.-board and the IC-board in Fig. 5c of the original manuscript has been corrected. The middle and bottom figures correspond to Fig. 5c in the original and revised manuscripts, respectively.

Comment #4:

Regarding lines 264-265 on page 15 stating 'To minimize the influence of the resistance variation on the transducer, the parallel memristor cell was chosen, which has been utilized in the memory and logic fields' - what specific advantages does this dual-memristor configuration offer compared to the previously mentioned single-memristor structure? How does this architectural improvement enhance the performance of your system?

Response: We sincerely appreciate the reviewer's valuable comments. As suggested, we have clarified the advantages of the dual-memristor (hereafter referred to as parallel two-memristor) configuration and its architectural benefits at biosensor system level. Our detailed explanations and supporting results are provided below.

The proposed biosensor operates in sequential sensing and reading stages. During the sensing stage, the memristor resistance (R_M) is set to either a high-resistance state resistance (R_{HRS}) or a low-resistance state resistance (R_{LRS}). In the subsequent reading stage, the output voltage (V_{OUTR}), determined by voltage dividing between the memristor resistance (R_M) and the output resistance (R_{OUT}), corresponds to a logic-low level when $R_M = R_{HRS}$ and to a logic-high level when $R_M = R_{LRS}$, as shown in Fig. 1e. Therefore, variations in R_M directly propagate to variations in V_{OUTR} ; consequently, smaller R_M variation leads to smaller V_{OUTR} variation. In this regard, the parallel two-memristor cell was employed in this work to reduce R_M variation and thereby reduce the variation of V_{OUTR} , meaning that the biosensor can operate reliably even in the presence of device variability and environmental noise.

The impact of a parallel two-memristor cell on V_{OUTR} was quantitatively assessed. The S5 species exhibited degraded resistance variation of R_{HRS} increased from 15.2% to 19.5% and that of R_{LRS} increased from 4.5% to 8.5% due to the absence of an ESD protection during integration into the biosensor and the use of wire bonding. Using these degraded S5 devices, V_{OUTR} of the transducers was simulated with the LTspice simulator (see Figure R2). As determined from the simulation results, the biosensors based on the single-memristor cell (S-transducer) and the parallel two-memristor cell (P-transducer) exhibited voltage swings between the logic-high and logic-low levels of 140 mV and 120 mV, respectively, both of which satisfied the 100 mV criterion set in this work to verify the proper operation of the threshold-sensing function. The variation of V_{OUTR} at logic-high and logic-low levels decreased from 0.52% and 3.9% with the S-transducer to 0.21% and 2.1% with the P-transducer, respectively. This indicates that the P-transducer exhibited approximately 50% lower V_{OUTR} variation compared with S-transducer, which means a more stable transducer operation.

Figure R2. The effect of the parallel two-memristor cell configuration on the biosensor transducer. **a.** I-V curve fitting results for measured and modeled memristors, showing good agreement in the positive voltage region, which directly contributes to the threshold operation of the transducer. **b-c.** Simulated transient characteristic for the output voltage in the reading stage (V_{OUTR}) of transducers based on a single-memristor cell (S-transducer) and a parallel two-memristor cell (P-transducer). The S-transducer and P-transducer exhibited voltage swings between the logic-high and logic-low levels of 140 mV and 120 mV, respectively, both of which satisfied the 100 mV criterion set in this work to verify the proper operation of the threshold-sensing function. The variation of V_{OUTR} at logic-high and logic-low levels decreased from 0.52% and 3.9% with the S-transducer to 0.21% and 2.1% with the P-transducer, respectively. This indicates that the P-transducer exhibited approximately 50% lower V_{OUTR} variation compared with S-transducer, which means a more stable transducer operation.

To address the reviewer's comments, we have appended Figure R2 as Supplementary Figure S6 in the revised Supplementary Information. The appended explanations in the revised manuscripts are as follows:

Page 15, line 258: The parallel memristor cell, previously adopted in memory and logic applications³⁴, was employed to minimize the influence of resistance variation of the transducer, thereby reducing the variation of V_{OUTR} , which means a more stable transducer operation (see Supplementary Fig. S6).

Comment #5:

Line 387 on page 18 mentions that 'the simplified biosensor version using a comparator, hold circuit, and LED to replace bulky ADC and display components could still accurately detect threshold points at pH 1 intervals.' However, the more complex version with ADC and MCU also achieved threshold detection at pH 1 intervals. Could the complex version potentially achieve higher detection precision? What are the advantages of the simplified version over the complex one? What is the performance gap between these two versions? We recommend adding comparative experiments.

Response: We sincerely appreciate your careful review of the device applications. As the reviewer mentioned, the comparison of the two demonstration versions is essential to highlight the advancements of this work. To address the reviewer's comments, we have appended the results of comparative experiments. There is no difference in detection precision between the complex version (hereafter referred to as the fully-integrated version) and the simplified version; however, there are performance differences in power dissipation and system size. The detailed explanations and results are shown below.

Both the fully-integrated version (including an ADC and a display) and the simplified version employ the same device, and the functional set/reset operation mechanism remains unchanged; accordingly, they exhibit identical detection precision. The distinction between the two versions lies in how the memristor state is read. In the fully-integrated version, this determination is made using an ADC, whereas in the simplified version it is made using a comparator. For user feedback, the fully-integrated version utilizes a display interfaced via

serial communication, while the simplified version indicates the comparator output with an LED. Although ADCs and serial communication are commonplace and straightforward functions, they become excessive for simply reading the binary state of the memristor cell and, as such, entail additional power consumption.

We have compared the experimental results of the two versions, as summarized in Table R1. During a 10-second sequence (set, init., reset, init.) for system operation, the average dissipated currents were measured to be 17.95 mA and 1.45 mA for the fully-integrated and simplified versions, respectively. The active areas of the IC boards were measured to be $60 \times 60 \text{ mm}^2$ and $30 \times 30 \text{ mm}^2$ for the fully-integrated and simplified versions, respectively.

The high power consumption of the fully-integrated version is likely to pose a significant challenge when transitioning beyond single-device demonstrations to integrated applications. For these reasons, we have included a simplified version that excludes ancillary peripherals such as the ADC and I²C (inter-integrated circuit for serial communication), retaining only essential functions. We anticipate that this approach will be particularly meaningful for applications in ultra-miniaturized sensor systems, where low-power operation is critical.

	Fully-integrated biosensor system	Simplified biosensor system
Used peripheral	GPIO, ADC, I ² C	GPIO
Method of sensing the memristor cell	ADC	Comparator
Method of user feedback	OLED display	LED
Tested pH range	2 - 10	2 - 10
Tested pH interval	1 pH	1 pH
The number of threshold pH point	1	1
Power consumption	17.95 mA	1.45 mA
Active size of IC board	$60 \times 60 \text{ mm}^2$	$30 \times 30 \text{ mm}^2$

Table R1. Experimental comparison between the fully-integrated and simplified versions of the biosensor systems. The simplified system achieved approximately a 4-fold reduction in IC board size and a 12.4-fold reduction in power consumption, relative to the fully-integrated system. Such a simplified system design is expected to be utilized for biosensor applications that require ultra-miniaturization and ultra-low power operation, such as ingestible sensor capsules.

To address the reviewer's comments and compared the two demonstrated systems, we have appended Table R1 as Supplementary Table S1 in the revised Supplementary Information. The appended explanations in the revised manuscripts are as follows:

Page 15, line 266: A prototype of the **fully-integrated** biosensor system was implemented, ~

Page 18, line 331: **The simplified system achieved approximately a 4-fold reduction in IC board size and a 12.4-fold reduction in power consumption, relative to the fully-integrated version (Supplementary Table S1).**

Comment #6:

Could you please supplement an experiment to demonstrate the real-time detection capability of the complex version (with ADC and MCU)?

Response: We sincerely appreciate the reviewer for providing the insightful suggestion to help further improve the quality of our work.

An experimental result has been provided as Figure R3 and Supplementary Video S1 to demonstrate the real-time detection capability of the complex version with ADC and MCU (hereafter referred to as the “fully-integrated system”). Figure R3 shows the experimental setup, and Supplementary Video S1 provides the real-time demonstration of the fully-integrated system. These results confirm that the implemented fully-integrated system successfully performs threshold operation for pH 2-10 solutions, as shown in Fig. 6d with a threshold point at pH 5.

To address the reviewer's comment, we have appended Figure R3 as Supplementary Figure S8 and Supplementary Video S1 in the revised Supplementary Information. The appended explanations in the manuscript are as follows:

Page 17, line 300: ~ **Supplementary Fig. S8, and Supplementary Video 1**). As the pH ~

Figure R3. Experimental setup of the demonstrated memristor-based fully integrated electrochemical biosensor system for pH sensing. The real-time operation of the system is shown in Supplementary Video 1, confirming successful threshold operation of pH 2-10 solutions, as presented in Fig. 6d with a threshold pH point at pH 5.

Comment #7:

Is the memristor strictly necessary for this work? Could other devices potentially replace the memristor's functionality, given that memristor chip fabrication is relatively complex?

Response: We sincerely thank the reviewer for this insightful question. As the reviewer pointed out, it is essential to provide a systematic explanation of why the memristor device is indispensable in the proposed biosensor transducer. Our detailed explanations are as follows.

The proposed biosensor transducer requires three key features: intrinsic threshold-sensing (TS) functionality enabling self-diagnosis of the analyte, non-volatile memory with long-term retention, and CMOS compatibility for monolithic integration with subsequent signal-processing blocks. However, existing devices cannot simultaneously provide all of these characteristics.

For example, DRAM is CMOS-compatible but volatile and NAND flash offers non-volatility with long-term retention but lacks TS functionality. Inorganic devices such as memristors, nanowires, and field-effect transistors may provide CMOS compatibility, but they lack intrinsic TS functionality¹⁻⁸. Organic devices, including nanowires, memristors, and transistors, have rarely been reported with TS functionality; nonetheless, they generally lack both long-term retention non-volatility and CMOS compatibility⁹⁻¹³.

In contrast, our memristor integrates these key requirements: intrinsic TS capability arising from its inherent single-level cell (SLC) behavior, non-volatility with decades-long retention (see Figure R4), and fully CMOS compatibility enabled by the TiN/HfO₂/TaO_x/TiN stack and the CMOS back-end-of-line (BEOL) process. A comparative summary with previously reported semiconductor device-based biosensors is provided in Table R2.

Figure R4. Retention performance of the fabricated memristor, demonstrating reliable non-volatile behavior with an extrapolated retention time of 10 years at approximately 431.3 K. The retention was extracted by tracking the failure time at elevated temperatures (573, 558, and 543 K) and extrapolating the fitted data to lower temperatures.

Ref	Type of Device	CMOS compatibility	Device Retention	Device On/off ratio	TS in transducer	Application	Demo. function	Demo. level	Sensor sensitivity	Sensor linearity
1	In-organic nano-wire	No	No	No	No	Electrochemical biosensor	PSA	Device	N/A	N/A
2	In-organic nano-wire	No	No	No	No		DNA	Device	N/A	N/A
3	In-organic graphene transistor	No	No	No	No		Heart failure	Device	N/A	99.17 %
4	In-organic EG-FET	Yes	No	No	No		Influenza virus	Device	26.7 mV/dec	N/A
5	In-organic memristor (Pt/Al ₂ O ₃ /TiO ₂ /Pt)	No	N/A	N/A	No		PSA	Device	N/A	N/A
6	In-organic memristor (IrO _x /GdO _x /W)	No	N/A	~1000	No		pH & Urea	Device	53.2 mV/pH	98.98%
7	In-organic memristor (Pt/NbTiO _x /NbTi)	No	N/A	1~5	No		pH	Device	N/A	98.0 %
8	In-organic ISFET	Yes	N/A	No	No		pH	Device	53.98 mV/pH	N/A
9	Organic nanowire (PU/AuNPs)	No	No	No	No		pH	System	58.9 mV/pH	99.99 %
10	Organic memristor (Au/MSFP/Au)	No	Short-term (2.7 h)	10	No	Neuromorphic in-sensor comput. system (training & classification)	Image recognition	System	-	-
11	Organic memristor (PEDOT:PSS)	No	Short-term (1 h)	6	No		Genetic disease	System	-	-
12	Organic memristor (PEDOT:PSS)	No	Short-term (1 s)	No	No		gesture recognition	System	-	-
13	Organic transistor (Poly)	No	Short-term (1 s)	200	Yes	Neuromorphic neuron	Neuronal behavior	Device	-	-
This Work	In-organic memristor (TiN/TaO _x /Ta ₂ O ₅)	Yes	Long-term (≥10 y)	35	Yes	Electrochemical biosensor	pH, Glucose, Ascorbic acid	System	57 mV/pH	98.1 %

Table R2. Comparison with previously reported semiconductor-based biosensors. Our biosensor represents the first semiconductor device-based electrochemical biosensor that demonstrates threshold-sensing (TS) functionality in the transducer level together with complete system-level integration. Our memristor offers full CMOS compatibility, which is readily extendable to monolithic integration with SP blocks, facilitating the realization of miniaturized sensor systems. Moreover, our transducer is the only one that simultaneously exhibits intrinsic TS functionality and long-term non-volatility (Fig. R4), which is valuable for miniaturized sensors with self-diagnosis capability.

To address the reviewer's comment, we have appended Figure R4 and Table R2 as Supplementary Figure S14 and Supplementary Table S2 in the revised Supplementary Information, respectively. The corresponding explanations in the revised manuscript are as follows:

Page 18, line 333: Our sensor technology is compared with previously reported semiconductor-based sensors (see Supplementary Table S2). To the best of our knowledge, the sensors developed in this work are the first demonstration of a semiconductor device-based electrochemical biosensor that exhibits TS functionality at the transducer level together with complete system-level integration. Unlike prior transducing devices, our memristor device also offers full CMOS compatibility, which is readily extendable to monolithic integration with SP blocks and thereby facilitating the realization of miniaturized sensor systems. Moreover, our transducer is the only one that simultaneously exhibits intrinsic TS functionality and long-term non-volatility (Supplementary Fig. S14), which is potentially valuable for miniaturized sensors with self-diagnosis capability requiring data retention.

Reviewer #2

Comment #1:

The application of sensing effect based on memristor has been reported in previous studies, and the author should highlight the innovation of this work by comparing it with the results reported previously.

Comment #4:

What are the important and unique applications of this work compared to other related reports (such as: Nature Communications, 2023, 14, 8489)? The author should clarify in the main text.

Response: We deeply appreciate the reviewer for these constructive comments. Since comments #1 and #4 above are closely related, we provide a combined response. As the reviewer mentioned, some memristor devices have been reported to operate sensors. Therefore, a detailed comparison is essential to highlight the advances of the proposed memristor-based sensor and it is necessary to clearly present the unique applications of this work. To address the reviewer's comments, we have compared our proposed sensor with previously reported ones, including the paper (Nature Communications, 2023, 14, 8489), and outlined its appropriate applications. The detailed explanations and results are provided below.

The results of these comparisons are shown in Table R2 like below. As summarized in Table R2, several memristor devices have been utilized to demonstrate various sensors such as neuromorphic in-sensor computing systems for biomedical applications, artificial biological neuron, and electrochemical biosensors.

Ref	Type of Device	CMOS compatibility	Device Retention	Device On/off ratio	TS in transducer	Application	Demo. function	Demo. level	Sensor sensitivity	Sensor linearity
1	In-organic nano-wire	No	No	No	No	Electrochemical bio-sensor	PSA	Device	N/A	N/A
2	In-organic nano-wire	No	No	No	No		DNA	Device	N/A	N/A
3	In-organic graphene transistor	No	No	No	No		Heart failure	Device	N/A	99.17 %
4	In-organic EG-FET	Yes	No	No	No		Influenza virus	Device	26.7 mV/dec	N/A
5	In-organic memristor (Pt/Al ₂ O ₃ /TiO ₂ /Pt)	No	N/A	N/A	No		PSA	Device	N/A	N/A
6	In-organic memristor (IrO _x /GdO _x /W)	No	N/A	~1000	No		pH & Urea	Device	53.2 mV/pH	98.98%
7	In-organic memristor (Pt/NbTiO _x /NbTi)	No	N/A	1~5	No		pH	Device	N/A	98.0 %
8	In-organic ISFET	Yes	N/A	No	No		pH	Device	53.98 mV/pH	N/A
9	Organic nanowire (PU/AuNPs)	No	No	No	No		pH	System	58.9 mV/pH	99.99 %
10	Organic memristor (Au/MSFP/Au)	No	Short-term (2.7 h)	10	No	Neuromorphic in-sensor comput. system (training & classification)	Image recognition	System	-	-
11	Organic memristor (PEDOT:PSS)	No	Short-term (1 h)	6	No		Genetic disease	System	-	-
12	Organic mem-transistor (PEDOT:PSS)	No	Short-term (1 s)	No	No		gesture recognition	System	-	-
13	Organic transistor (Poly)	No	Short-term (1 s)	200	Yes	Neuromorphic neuron	Neuronal behavior	Device	-	-
This Work	In-organic memristor (TiN/TaO _x /Ta ₂ O ₅)	Yes	Long-term (≥10 y)	35	Yes	Electrochemical bio-sensor	pH, Glucose, Ascorbic acid	System	57 mV/pH	98.1 %

Table R2. Comparison with previously reported semiconductor-based biosensors. Our biosensor represents the first semiconductor device-based electrochemical biosensor that demonstrates threshold-sensing (TS) functionality in the transducer level together with complete system-level integration. Our memristor offers full CMOS compatibility, which is readily extendable to monolithic integration with SP blocks, facilitating the realization of miniaturized sensor systems. Moreover, our transducer is the only one that simultaneously exhibits intrinsic TS functionality and long-term non-volatility (Fig. R4), which is valuable for miniaturized sensors with self-diagnosis capability.

For the implementation of memristor-based sensor systems in miniaturized applications, threshold-sensing (TS) functionality, non-volatile memory with long-term retention, and CMOS compatibility are required in the transducer.

Memristor devices based on organic materials such as SFP¹⁰, PEDOT:PSS¹¹, and poly (benzimidazobenzophenanthroline)¹² have been developed for neuromorphic in-sensor computing applications, including image recognition¹⁰, genetic disease diagnosis¹¹, and wearable gesture recognition using EMG signals¹². The lack of analysis regarding CMOS-compatibility limits their utilization in reliable sensor applications. Moreover, their transducers for optical-to-electrical¹⁰, electrolyte-to-electrical¹¹, stimuli-to-electrical¹² conversion lack intrinsic TS functionality; instead, thresholding is relegated at the subsequent signal-processing stage, which poses a barrier to sensor miniaturization¹⁰⁻¹².

Another study has reported an organic electrochemical memtransistor (OECmT) that emulates an artificial neuron¹³. Although the device experimentally demonstrated intrinsic TS functionality, it not only lacks CMOS compatibility but also exhibits non-volatile memory behavior with a retention time of only a few seconds. This short-term retention property renders the device unsuitable for ultra-miniaturized electrochemical biosensors that demand reliable data retention even after power is turned off due to limited battery capacity.

Inorganic memristors have been developed for application in electrochemical biosensors⁵⁻⁷. However, their potential for reliable sensor applications is restricted by their CMOS incompatibility due to the use of Pt^{5,7}, GdO_x⁶, Nb⁷ materials. These memristors also lack intrinsic TS functionality, thereby increasing the burden on the subsequent signal-processing block and ultimately hindering the realization of ultra-miniaturized biosensors. Furthermore, their demonstrations have remained at the device level, lacking integration of signal-processing and display units as well as any circuit topology designed to tolerate device variation.

In contrast, our memristor integrates these key requirements for the first time: intrinsic TS functionality arising from its inherent single-level cell (SLC) behavior, non-volatile memory with decades-long retention (see Figure R4), and fully CMOS compatibility enabled by the TiN/HfO₂/TaO_x/TiN stack and the CMOS back-end-of-line (BEOL) process.

Figure R4. Retention performance of the fabricated memristor, demonstrating reliable non-volatile behavior with an extrapolated retention time of 10 years at approximately 431.3 K. The retention was extracted by tracking the failure time at elevated temperatures (573, 558, and 543 K) and extrapolating the fitted data to lower temperatures.

The key innovations of this study are as follows.

1) The sensors developed in this work are the **first demonstration** of a semiconductor device-based electrochemical biosensor that exhibits **TS functionality at the transducer level together with complete system-level integration**, to the best of our knowledge. Achieving this level of performance is particularly remarkable in filamentary memristors, which inherently exhibit stochastic behavior.

2) Unlike most of prior transducing devices^{1-3, 5-7, 9-13}, our memristor device also offers **full CMOS compatibility**, which is readily extendable to monolithic integration with SP blocks and can thereby facilitate the realization of miniaturized sensor systems.

3) **Our transducer presents the first demonstration of TS functionality together with long-term non-volatility at the transducer level.** Previously reported memristors have relied on multi-level cell (MLC) characteristics to perform analog sensing⁵⁻⁷. By necessity, the discrimination of the sensed electrical signals must be carried out in subsequent signal-processing stages or external computers, which can render the overall system bulky and energy-inefficient due to the use of either a complex signal-processing unit or wired/wireless communication. Unlike these conventional memristor-based analog sensing approaches that exploit MLC characteristics, our approach uniquely utilizes threshold sensing via single-level cell (SLC) characteristics. Our TiN/HfO₂/TaOx/TiN memristor ensures sufficient read margin due to a well-defined resistance separation between HRS and LRS states, which we leveraged to realize, for the first time, a threshold-sensing-based sensor transducer. Although TS functionality at the device level has been previously reported in an organic transistor¹³, that

device is incompatible with CMOS processes and exhibits a retention time of only ~ 1 s. Our transducer is the only one that simultaneously exhibits intrinsic TS functionality and non-volatility with long-term retention, which is potentially valuable for miniaturized sensors with self-diagnosis capability requiring data retention even under power-off conditions due to limited battery capacity.

4) This work presents the first demonstration of resistance-variation tolerant transducer topology. No prior memristor-based biosensors have resolved the memristor variation from a circuit-design perspective. Conventional memristor-based sensors directly read the memristor resistance (R_M), so their output waveform inevitably reflected this variation⁵⁻⁷. In this work, we address the resistance variation problem by introducing a resistance-parallelization scheme. As shown in Fig. R5a, the parallel memristor cell resistance (R_{MP}) is connected in parallel with the relatively small on-resistance (R_T) of the integrated transistor, yielding an equivalent resistance R_{EQ} that is tolerant to R_M variation when R_T is designed to be much smaller than R_M ($R_T \ll R_M$). At the sensing stage, R_{EQ} is multiplied by the transistor drain current (I_D) to generate the transducer output voltage (V_{OUTS}). As determined from our LTspice simulation results, when the R_M variation was 19.2% as obtained in this work, the R_{EQ} exhibited less than 1% variation coefficient, and consequently V_{OUTS} also showed less than 1% variation coefficient (see Figure R5b). This variation-immune V_{OUTS} enabled the implemented pH sensor system to exhibit the same threshold pH point within a one-pH interval (see Figures R5c-R5d and Figure 6d in the revised manuscript).

Figure R5. The effect of the resistance parallelization scheme on the biosensor transducer. a. Circuit diagram of the biosensor transducer at the sensing stage. **b.** Simulated parallel resistance (R_{EQ}) and sensing-stage output voltage (V_{OUTS}) characteristics based on the measured R_{HRS} (HRS resistance of memristors) variation of 19.2%. The R_{EQ} variation coefficient was considerably reduced to less than 1%, resulting in a V_{OUTS} variation coefficient also below 1%. **c.** Simulated switching characteristics of the parallel memristor cell resistance (R_{MP}) under increasing V_{OUTS} with a variation coefficient of less than 1%. With V_{OUTS} increased in ~ 26 mV step, R_{MP} abruptly switched from $\sim 700 \Omega$ to $\sim 80 \Omega$ at about 700 mV, indicating the presence of only one threshold pH point (TP). **d.** Simulated number of TP as a function of the V_{OUTS} variation coefficient, confirming that V_{OUTS} variation of less than 1% ensures only one TP. The simulations in b-d were performed using 10 measured memristor devices, and the set-voltage variation of the devices was neglected to examine the pure effect of V_{OUTS} variation.

The specific application for this work is presented as follows.

This study is not aimed at neuromorphic computing applications¹⁰⁻¹², including Nature Communications (2023, 14, 8489)¹⁰, where memristor crossbar arrays are employed to learn and infer analytes. Neuromorphic computing targets ultra-low energy consumption compared with conventional GPUs, with requirements such as at least 100 billion memristors comparable to the number of human synapses, device dimensions below 10 nm, and energy consumption below 10 fJ per synaptic operation. Achieving such specifications still requires major technological breakthroughs.

This study focuses on endowing the transducer with threshold-diagnosis capability and long-term storage of the diagnostic information. A typical sensor consists of sensing electrodes, a transducer, a signal-processing block, a communication block, and a computer or display. When the transducer itself has threshold-diagnosis and non-volatile retention capabilities, the subsequent signal-processing block can be considerably simplified through the elimination of ADCs, comparators, and MCUs. Moreover, because diagnosis is performed

within the sensor, real-time external communication is no longer required, allowing removal of the communication block. Furthermore, because the diagnostic state is non-volatilely stored in the transducer, the sensor needs power only during the diagnosis period; once the decision is recorded, power can be turned off without information loss, thereby drastically reducing the required battery capacity.

Based on these characteristics, **our CMOS-compatible TS transducer technology with long-term non-volatile retention is highly suitable for the ultra-low-power operation and miniaturization of conventional electrochemical biosensors. A representative application is an ingestible sensor capsule** (see Figure R6). Conventional capsules¹⁴⁻¹⁶ rely on real-time wireless transmission to external receivers to diagnose diseases, forcing patients to wear a waist-mounted device. Their lengths typically exceed 2 cm, which can cause discomfort during swallowing and gastrointestinal transit. In contrast, our capsule eliminates real-time wireless transmission and external receivers by leveraging the transducer's intrinsic threshold-based diagnosis (see Figures R6a-R6c). The diagnosed information is stored non-volatilely in the transducer, enabling retrieval after power is turned off and considerably decreasing the required battery capacity. Because batteries typically occupy more than 50 % of capsule volume, the overall form factor can be reduced to less than half of that of conventional systems, i.e., down to 11.5 mm in length and 5.5 mm in diameter. Power modeling indicates 13.2 days of operation at a 1-minute sampling interval and 4.7 days at a 20-second interval, assuming an 8.66 mA draw for 16.8 ms during each set operation of the simplified circuit (see Figure R6d).

The capsule can integrate multiple memristor-based transducers preset to the characteristic pH thresholds of digestive organs (stomach 2.6, small intestine 7.4, large intestine 6.5¹⁴). When the local pH drops below its threshold, the corresponding device switches from a high-resistance state (HRS) to a low-resistance state (LRS), effectively recording the physiological state in situ. The capsule typically exits the body within ~20 hours; once retrieved, it can be re-powered to read out the stored resistance states, enabling physicians to retrospectively identify abnormal conditions encountered along the gastrointestinal tract.

While memristor technology is rapidly advancing in biosensing applications, its specific use in gastrointestinal pH monitoring devices remains largely unexplored, indicating a valuable opportunity for innovation in future studies.

Figure R6. Potential application of the memristor-based electrochemical biosensor transducer with threshold-sensing functionality. **a.** Exploded view of memristor integrated ingestible sensor capsule. **b.** Schematic illustration of ingestible sensor capsule including multimodal sensors for gastrointestinal tract studies^{17,18}. **c.** Block diagram of the sensor system. The ultra-low-leakage switch minimizes power loss during sleep mode. A nano-power system timer generates the clock signal for memristor operation and power switching, while a switch array manages the memristor array and its programming sequence. **d.** The estimated battery life equation and the estimated number of batteries when the measurement of the 20-second cycle is performed.

To highlight the advantages of the proposed memristor-based biosensor, we measured the retention characteristic of the fabricated memristors. The results showed that the memristor exhibits non-volatile memory with long-term retention of approximately 10 years at 431.3 K (see Figure R4), confirming its suitability for use in the electrochemical biosensor transducer with TS functionality. We further appended simulation results quantifying the effect of R_M variation on the transducer output voltage (V_{OUTS}), showing strong tolerance to R_M variation: a 19.2% variation in HRS translates to less than 1 % variation in V_{OUTS} , thereby yielding only one TP (see Figure R5). In addition, we appended a potential application for our biosensor with TS functionality as an ultra-miniaturized ingestible capsule (see Figure R6).

To address the reviewer’s comment, we have appended Figures R4, R5, and R6 and Table R2 as Supplementary Figures S14, S7, and S15 and Supplementary Table S2 in the revised Supplementary Information, respectively. The corresponding explanations in the revised

manuscript are as follows:

Page 17, line 291: Although the memristors exhibited device-to-device resistance variation, the resistance parallelization scheme, by employing variation-tolerant REQ instead of directly using RMP, effectively suppressed this variation and enabled a stable VOUTS response with a clearly defined resistance switching characteristic of the parallel memristor cell (see Supplementary Figures S5 and S7).

Page 18, line 333: Our sensor technology is compared with previously reported semiconductor-based sensors (see Supplementary Table S2). To the best of our knowledge, the sensors developed in this work are the first demonstration of a semiconductor device-based electrochemical biosensor that exhibits TS functionality at the transducer level together with complete system-level integration. Unlike prior transducing devices, our memristor device also offers full CMOS compatibility, which is readily extendable to monolithic integration with SP blocks and thereby facilitating the realization of miniaturized sensor systems. Moreover, our transducer is the only one that simultaneously exhibits intrinsic TS functionality and long-term non-volatility (Supplementary Fig. S14), which is potentially valuable for miniaturized sensors with self-diagnosis capability requiring data retention.

Page 19, line 342: Owing to its simplicity, scalability, and functionality, the proposed memristor-based technology holds strong potential for portable point-of-care diagnostic applications. Furthermore, in applications such as ingestible capsule sensors, this technology offers distinct advantages, such as miniaturization, non-volatile memory, scalability, and ultra-low power consumption, while addressing future challenges associated with safely monitoring internal physiological environments with minimal discomfort and risk to the user (Supplementary Fig. S15).

Page 20, line 374: Owing to its simplicity, scalability, and functionality, the proposed memristor-based technology holds strong potential not only for portable point-of-care diagnostic applications but also for miniaturized sensing devices that require inherent memory retention.

Comment #2:

The production and quality of figures need to be further improved.

Response: We sincerely appreciate the reviewer's valuable comment regarding the improvement of figure quality. In response, we have substantially **improved the production and quality of all figures in both the main manuscript and the Supplementary Information** to ensure high clarity and readability. Please kindly refer to the revised manuscript and Supplementary information.

For example, in Figure 5c shown below, we increased the figure resolution, corrected the incorrect wiring between the IC board and the mem.-board, and improved the readability of the text in the right panel.

One of previous figures (Figure 5c in the original manuscript)

One of revised figures (Figure 5c in the revised manuscript)

Figure R7. An example comparing figure production and quality before and after revision

Comment #3:

The formation of conductive filaments requires relevant experimental evidence, such as in situ TEM images.

Response: We deeply appreciate the reviewer for the valuable comments. As the reviewer mentioned, experimental evidence using advanced materials analysis is important to verify the proposed current switching mechanism.

Because conductive filaments are only 1–5 nm in diameter and randomly distributed in three dimensions within the switching layer, their direct observation was highly time-consuming and required substantial effort and expense. Our group has not been able to access a facility equipped with in situ TEM that provides sufficient measurement time.

As an alternative, we performed ex situ STEM-EELS (scanning transmission electron microscopy - electron energy loss spectroscopy). After inducing filament formation in a memristor on the S5 die by applying a SET voltage, we prepared a cross-section using focused ion beam (FIB) and scanned the switching layer with STEM. The total field of view was $\sim 100 \times 30 \text{ nm}^2$ with a pixel size of 5 nm (see Figure R7a).

Figure R7b shows EELS spectra acquired at six pixels (P1–P6) along a vertical line can: P1 (top TiN/TaO_x interface), P2–P4 (TaO_x), P5 (Ta₂O₅), and P6 (Ta₂O₅/bottom TiN interface). At P5, a sharp O-K edge feature at $\sim 533.5 \text{ eV}$ indicates stoichiometric Ta₂O₅¹⁹, whereas P2–P4 exhibit reduced peak intensity at $\sim 533.5 \text{ eV}$, consistent with oxygen-deficient TaO_x¹⁹.

The observation of stoichiometric Ta₂O₅ at P5, rather than the expected oxygen-deficient TaO_x associated with filament formation is believed to be inevitable, as it likely arises from the relatively large pixel size of STEM (5 nm) compared with the filament diameter (1–5 nm), and the limited lateral scan width (100 nm), which samples only $\sim 6 \%$ of the 1.7 μm -wide switching layer of the memristor.

As ex situ STEM-EELS measurements are also highly time-consuming, we regret that it is not feasible for us to address this comment within the current constraints of limited time. We note that the direct observation of conductive filaments is itself challenging enough to constitute a high-quality publication¹. We note that the direct observation of conductive filaments is sufficiently challenging that it could constitute a high-quality publication²⁰. In future work, we will decrease the pixel size, expand the scanning area, and pursue in situ STEM measurements. If successful, we would like to report them in a separate publication.

If further information or measurements on these points would be helpful, please let us know;

we will incorporate them promptly.

Figure R8. Ex-situ STEM-EELS scan of the fabricated TiN/TaO_x/Ta₂O₅/TiN memristor after filament formation. a. STEM cross-sectional image. The total scanning area was $100 \times 30 \text{ nm}^2$ with a pixel size of 5 nm. **b.** O-K edge EELS spectra of six pixels (P1–P6) along a vertical column: P1 (top TiN/TaO_x interface), P2–P4 (TaO_x), P5 (Ta₂O₅), and P6 (Ta₂O₅/bottom TiN interface).

Comment #5:

English grammar and writing errors require careful correction by the author.

Response: We sincerely appreciate the reviewer's constructive comments and valuable opportunity to further improve the quality of our manuscript. The English errors in the initial manuscript have been carefully and thoroughly corrected in the revised main manuscript and Supplementary Information, where all corrected parts are highlighted in red. The revised version is now free of typographical and grammatical errors. Representative examples of these corrections are provided below. We sincerely hope that these errors in the original manuscript will not detract from the significance of the research innovations and contributions we have achieved in this work. We would be truly grateful for your kind and considerate evaluation.

Page 2, line 52: (Grammar correction)

- Original manuscript: When the V_{REF} changes from 1.64 to 2.2 V, the threshold pH point (TP) is adjusted to any pH value between pH 2 to 10 in units of pH 1, exhibiting good linearity between the relation of TP and pH values with a Pearson's correlation coefficient of 0.981.
- Revised manuscript: **Furthermore, by varying V_{REF} from 1.64 to 2.2 V, the TP could be tuned to any desired pH value between pH 2 and 10 in increments of 1, demonstrating** good linearity between the relation of TP and pH values with a Pearson's correlation coefficient of 0.981.

Page 4, line 89: (writing correction)

- Original manuscript: Owing to the dense cell size of $4F^2$ and low set/reset voltage of less than 1V of the memristors, they have been used as cells in cross-bar arrays of neuromorphic systems and have improved power consumption and energy efficiency of the system^{12,13}.
- Revised manuscript: Owing to the dense cell size of $4F^2$ and low set/reset voltage of less than **1 V** of the memristors, they have been used as cells in cross-bar arrays of neuromorphic systems and have **reduced power consumption and improved energy efficiency of the system**^{12,13}.

Page 7, line 134: (Grammar correction)

- Original manuscript: The single-bit memristor features that R_M is changed from R_{HRS} (OR R_{LRS}) to R_{LRS} (OR R_{HRS}) by the SET (or RESET) process of applying a set voltage (V_{SET}) (or reset voltage (V_{RST})), and a forming process of applying a forming voltage (V_{FORM}) is required to initially form conductive filaments of the memristor^{22,23}

- Revised manuscript: The single-bit memristor is characterized by the change of R_M from R_{HRS} (or R_{LRS}) to R_{LRS} (or R_{HRS}) through the SET (or RESET) process by applying a set voltage (V_{SET}) (or reset voltage (V_{RST})), while a forming process with a forming voltage (V_{FORM}) is required to initially form conductive filaments of the memristor^{22,23}

Page 10, line 196: (writing correction)

- Original manuscript: ones of S1, as shown in Fig. S1, is a piece of evidence supporting ~

- Revised manuscript: ones of S1 (Supplementary Fig. S2) is a piece of evidence supporting

Page 10, line 198: (Grammar correction)

- Original manuscript: assuming that the same number of oxygen anions are contributed in filament rupture,

- Revised manuscript: assuming that the same number of oxygen anions contribute to filament rupture,

Page 10, line 205: (writing correction)

- Original manuscript: The O₂/Ar gas mixture ratio of TaO_x was split into 9 (S4), 6 (S5), and 3 % (S6).

- Revised manuscript: The O₂/Ar gas mixture ratio of TaO_x was varied at 9 (S4), 6 (S5), and 3 % (S6).

Page 11, line 211: (Grammar correction)

- Original manuscript: ~layers compared to the 40 nm-thick ones was smaller due to its physically thinner thickness.

- Revised manuscript: ~layers was smaller compared to the 40 nm-thick species due to their less thickness.

Page 11, line 214: (writing correction)

- Original manuscript: while achieving the same x mole fraction value of TaO_x of 0.54.

- Revised manuscript: while achieving the same x mole fraction value of TaO_x of 0.54.

Page 12, Fig.4 Caption: (writing correction)

- Original manuscript: **b.** Measured short-pulse program/erase(P/E) cyclings of the TaO_x/Ta₂O₅ memristor (S5) based on a 20 nm-thick TaO_x layer with an O₂/Ar gas mixture ratio of 6 % under a condition of a unit pulse time (T_{PULSE}) of 2 μ s, showing an on/off ratio of more than 35 until number of pulses of 30000. 100 consecutive SET pulses (1 V for 2 μ s) followed by 100 consecutive RESET pulses (-1.5 V for 2 μ s) for a P/E cycling are applied to the S5 memristor and the current value was

read by read pulses (3 V for 10 μ s). **c.** Measured long-pulse P/E cyclings of the TaO_x/Ta₂O₅ memristor (S5) under a condition of T_{PULSE} of 50 μ s, exhibiting an on/off ratio over 30 until the pulse numbers reaches 4000. 100 consecutive SET pulses (1 V for 50 μ s) followed by 100 consecutive RESET pulses (-1.5 V for 50 μ s) are applied to the S5 memristor and the current value is read by read pulses (3 V for 10 μ s).

- Revised manuscript: **b.** Measured long-pulse program/erase(P/E) cyclings of the TaO_x/Ta₂O₅ memristor (S5) based on a 20 nm-thick TaO_x layer with an O₂/Ar gas mixture ratio of 6 % under a condition of a unit pulse time (T_{PULSE}) of 50 μ s, exhibiting an on/off ratio over 30 until the pulse numbers reaches 4000. 100 consecutive SET pulses (1 V for 50 μ s) followed by 100 consecutive RESET pulses (-1.5 V for 50 μ s) are applied to the S5 memristor and the current value is read by read pulses (0.3 V for 10 μ s). **c.** Measured short-pulse P/E cyclings of the TaO_x/Ta₂O₅ memristor (S5) under a condition of T_{PULSE} of 2 μ s, showing an on/off ratio of more than 35 until number of pulses of 30000. 100 consecutive SET pulses (1 V for 2 μ s) followed by 100 consecutive RESET pulses (-1.5 V for 2 μ s) for a P/E cycling are applied to the S5 memristor and the current value was read by read pulses (0.3 V for 10 μ s).

Page 12, line 221: (stylistic editing)

- Original manuscript: Although some results have been reported previously on reducing the current of CMOS-incompatible memristors by lowering only the O₂/Ar ratio^{28,30} or reducing only the thickness of oxygen reservoir layers^{32,33}, this is the first time that an increase in on/off ratio as well as a decrease in the current of CMOS-compatible memristors have been achieved by simultaneously adjusting the O₂/Ar ratio and the thickness of the oxygen reservoir, to our knowledge.

- Revised manuscript: While previous reports achieved current reduction in CMOS-incompatible memristors by adjusting only the O₂/Ar ratio^{28,30} or only the oxygen reservoir thickness^{32,33}, our study is, to the best of our knowledge, the first to demonstrate both an enhanced on/off ratio and a reduced current in CMOS-compatible memristors by simultaneously tuning the O₂/Ar ratio and the oxygen reservoir thickness.

Page 13, line 238: (Grammar, writing correction)

- Original manuscript: exhibiting a measured on/off ratio value of more than 35 up to number of pulses of 30000 (Fig. 4c), which was a good value among the previously reported same CMOS process-compatible memristor stacks of TiN/TaO_x/Ta₂O₅/TiN that did not use noble metals such as Pt, Pt, Au as electrodes

- Revised manuscript: exhibiting a measured on/off ratio value of more than 35 up to **several** pulses of 30000 (Fig. 4c), which was a good value among the previously reported same CMOS process-

compatible memristor stacks of TiN/TaO_x/Ta₂O₅/TiN that did not use noble metals such as Pt, Au as electrodes

Page 15, line 258: (stylistic editing)

- Original manuscript: To minimize the influence of the resistance variation on the transducer, the parallel memristor cell was chosen, which has been utilized in the memory and logic fields

- Revised manuscript: The parallel memristor cell, previously adopted in memory and logic applications³⁴, was employed to minimize the influence of resistance variation of the transducer, thereby reducing the variation of V_{OUTR} , which means a more stable transducer operation (see Supplementary Fig. S6).

Supplementary Fig. S1 Caption: (writing correction)

- Original manuscript: TaO_x/Ta₂O₅ in the fabricated memristor species of S2 to S4.

- Revised manuscript: TaO_x/Ta₂O₅ in the fabricated memristor species of S1 to S3.

Reviewer #3

This paper reports on a TaOx/Ta₂O₅ memristor device designed for biomarker threshold-sensing applications. The authors conducted extensive research on memristor performance and demonstrated the concept by integrating a sensor electrode for pH sensing, a signal processing (SP) module, and a display with the proposed transducer. However, the pH demonstration falls short of illustrating the biomarker detection capabilities mentioned in the introduction, such as C-reactive protein detection, which presents significantly greater technical challenges for accurate measurement. The authors' contributions primarily focus on memristor characterization.

Comment #0:

Given that TaOx/Ta₂O₅ materials have been widely applied in memristors and numerous reports on biomarker threshold detection already exist (Nature Electronics, 2023, 6, 765–770; Nature Electronics, 2024, 7, 1176–1185; Nature Communications, 2025, 16, 4334, among others), the novelty presented in this paper is insufficient for publication in Nature Communications.

Response: We deeply appreciate the reviewer for these constructive comments. As the reviewer mentioned, numerous devices have been reported to operate sensors. Therefore, a detailed comparison is essential to highlight the advances of the proposed memristor-based sensor. To address the reviewer's comments, we have compared our proposed sensor with previously reported ones, including the papers (Nature Electronics, 2023, 6, 765–770; Nature Electronics, 2024, 7, 1176–1185; Nature Communications, 2025, 16, 4334). The detailed explanations and results are provided below.

The results of these comparisons are shown in Table R2 like below (Refs. 11, 12, and 13 correspond to *Nat. Electron.* (2023, 6, 765–770), *Nat. Electron.* (2024, 7, 1176–1185), and *Nat. Commun.* (2025, 16, 4334), respectively). As summarized in Table R2, several devices have been utilized to demonstrate various sensors such as neuromorphic in-sensor computing systems for biomedical applications, neuromorphic biological neuron, and electrochemical biosensors.

Ref	Type of Device	CMOS compatibility	Device Retention	Device On/off ratio	TS in transducer	Application	Demo. function	Demo. level	Sensor sensitivity	Sensor linearity
1	In-organic nano-wire	No	No	No	No	Electrochemical bio-sensor	PSA	Device	N/A	N/A
2	In-organic nano-wire	No	No	No	No		DNA	Device	N/A	N/A
3	In-organic graphene transistor	No	No	No	No		Heart failure	Device	N/A	99.17 %
4	In-organic EG-FET	Yes	No	No	No		Influenza virus	Device	26.7 mV/dec	N/A
5	In-organic memristor (Pt/Al ₂ O ₃ /TiO ₂ /Pt)	No	N/A	N/A	No		PSA	Device	N/A	N/A
6	In-organic memristor (IrO _x /GdO _x /W)	No	N/A	~1000	No		pH & Urea	Device	53.2 mV/pH	98.98%
7	In-organic memristor (Pt/NbTiO _x /NbTi)	No	N/A	1~5	No		pH	Device	N/A	98.0 %
8	In-organic ISFET	Yes	N/A	No	No		pH	Device	53.98 mV/pH	N/A
9	Organic nanowire (PU/AuNPs)	No	No	No	No		pH	System	58.9 mV/pH	99.99 %
10	Organic memristor (Au/MSFP/Au)	No	Short-term (2.7 h)	10	No	Neuromorphic in-sensor comput. system (training & classification)	Image recognition	System	-	-
11	Organic memristor (PEDOT:PSS)	No	Short-term (1 h)	6	No		Genetic disease	System	-	-
12	Organic mem-transistor (PEDOT:PSS)	No	Short-term (1 s)	No	No		gesture recognition	System	-	-
13	Organic transistor (Poly)	No	Short-term (1 s)	200	Yes	Neuromorphic neuron	Neuronal behavior	Device	-	-
This Work	In-organic memristor (TiN/TaO _x /Ta ₂ O ₅)	Yes	Long-term (≥10 y)	35	Yes	Electrochemical bio-sensor	pH, Glucose, Ascorbic acid	System	57 mV/pH	98.1 %

Table R2. Comparison with previously reported semiconductor-based biosensors. Our biosensor represents the first semiconductor device-based electrochemical biosensor that demonstrates threshold-sensing (TS) functionality in the transducer level together with complete system-level integration. Our memristor offers full CMOS compatibility, which is readily extendable to monolithic integration with SP blocks, facilitating the realization of miniaturized sensor systems. Moreover, our transducer is the only one that simultaneously exhibits intrinsic TS functionality and long-term non-volatility (Fig. R4), which is valuable for miniaturized sensors with self-diagnosis capability.

For the implementation of memristor-based sensor systems in miniaturized applications, threshold-sensing (TS) functionality, non-volatile memory with long-term retention, and CMOS compatibility are required in the transducer.

Memristor devices based on organic materials such as SFP¹⁰, PEDOT:PSS¹¹, and poly (benzimidazobenzophenanthroline)¹² have been developed for neuromorphic in-sensor computing applications, including image recognition¹⁰, genetic disease diagnosis¹¹, and wearable gesture recognition using EMG signals¹². The lack of analysis regarding CMOS-compatibility limits their utilization in reliable sensor applications. Moreover, their transducers for optical-to-electrical¹⁰, electrolyte-to-electrical¹¹, stimuli-to-electrical¹² conversion lack intrinsic TS functionality; instead, thresholding is relegated at the subsequent signal-processing stage, which poses a barrier to sensor miniaturization¹⁰⁻¹².

Another study has reported an organic electrochemical memtransistor (OECmT) that emulates an artificial neuron¹³. Although the device experimentally demonstrated intrinsic TS functionality, it not only lacks CMOS compatibility but also exhibits non-volatile memory behavior with a retention time of only a few seconds. This short-term retention property renders the device unsuitable for ultra-miniaturized electrochemical biosensors that demand reliable data retention even after power is turned off due to limited battery capacity.

Inorganic memristors have been developed for application in electrochemical biosensors⁵⁻⁷. However, their potential for reliable sensor applications is restricted by their CMOS incompatibility due to the use of Pt^{5,7}, GdO_x⁶, Nb⁷ materials. These memristors also lack intrinsic TS functionality, thereby increasing the burden on the subsequent signal-processing block and ultimately hindering the realization of ultra-miniaturized biosensors. Furthermore, their demonstrations have remained at the device level, lacking integration of signal-processing and display units as well as any circuit topology designed to tolerate device variation.

In contrast, our memristor integrates these key requirements for the first time: intrinsic TS functionality arising from its inherent single-level cell (SLC) behavior, non-volatile memory with decades-long retention (see Figure R4), and fully CMOS compatibility enabled by the TiN/HfO₂/TaO_x/TiN stack and the CMOS back-end-of-line (BEOL) process.

Figure R4. Retention performance of the fabricated memristor, demonstrating reliable non-volatile behavior with an extrapolated retention time of 10 years at approximately 431.3 K. The retention was extracted by tracking the failure time at elevated temperatures (573, 558, and 543 K) and extrapolating the fitted data to lower temperatures.

The key innovations of this study are as follows. 1) The sensors developed in this work are the **first demonstration** of a semiconductor device-based electrochemical biosensor that exhibits **TS functionality at the transducer level together with complete system-level integration**, to the best of our knowledge. Achieving this level of performance is particularly remarkable in filamentary memristors, which inherently exhibit stochastic behavior.

2) Unlike most of prior transducing devices^{1-3, 5-7, 9-13}, our memristor device also offers **full CMOS compatibility**, which is readily extendable to monolithic integration with SP blocks and can thereby facilitate the realization of miniaturized sensor systems.

3) **Our transducer presents the first demonstration of TS functionality together with long-term non-volatility at the transducer level.** Previously reported memristors have relied on multi-level cell (MLC) characteristics to perform analog sensing⁵⁻⁷. By necessity, the discrimination of the sensed electrical signals must be carried out in subsequent signal-processing stages or external computers, which can render the overall system bulky and energy-inefficient due to the use of either a complex signal-processing unit or wired/wireless communication. Unlike these conventional memristor-based analog sensing approaches that exploit MLC characteristics, our approach uniquely utilizes threshold sensing via single-level cell (SLC) characteristics. Our TiN/HfO₂/TaOx/TiN memristor ensures sufficient read margin due to a well-defined resistance separation between HRS and LRS states, which we leveraged to realize, for the first time, a threshold-sensing-based sensor transducer. Although TS functionality at the device level has been previously reported in an organic transistor¹³, that

device is incompatible with CMOS processes and exhibits a retention time of only ~ 1 s. Our transducer is the only one that simultaneously exhibits intrinsic TS functionality and long-term non-volatility, which is potentially valuable for miniaturized sensors with self-diagnosis capability requiring data retention even under power-off conditions due to limited battery capacity.

4) This work presents the first demonstration of resistance-variation tolerant transducer topology. No prior memristor-based biosensors have resolved the memristor variation from a circuit-design perspective. Conventional memristor-based sensors directly read the memristor resistance (R_M), so their output waveform inevitably reflected this variation⁵⁻⁷. In this work, we address the resistance variation problem by introducing a resistance-parallelization scheme. As shown in Fig. R5a, the parallel memristor cell resistance (R_{MP}) is connected in parallel with the relatively small on-resistance (R_T) of the integrated transistor, yielding an equivalent resistance R_{EQ} that is tolerant to R_M variation when R_T is designed to be much smaller than R_M ($R_T \ll R_M$). At the sensing stage, R_{EQ} is multiplied by the transistor drain current (I_D) to generate the transducer output voltage (V_{OUTS}). As determined from our LTspice simulation results, when the R_M variation was 19.2 % as obtained in this work, the R_{EQ} exhibited less than 1 % variation coefficient, and consequently V_{OUTS} also showed less than 1 % variation coefficient (see Figure R5b). This variation-immune V_{OUTS} enabled the implemented pH sensor system to exhibit the same threshold pH point within a one-pH interval (see Figures R5c-R5d and Figure 6d in the revised manuscript).

Figure R5. The effect of the resistance parallelization scheme on the biosensor transducer. a. Circuit diagram of the biosensor transducer at the sensing stage. **b.** Simulated parallel resistance (R_{EQ}) and sensing-stage output voltage (V_{OUTS}) characteristics based on the measured R_{HRS} (HRS resistance of memristors) variation of 19.2%. The R_{EQ} variation coefficient was considerably reduced to less than 1%, resulting in a V_{OUTS} variation coefficient also below 1%. **c.** Simulated switching characteristics of the parallel memristor cell resistance (R_{MP}) under increasing V_{OUTS} with a variation coefficient of less than 1%. With V_{OUTS} increased in ~ 26 mV step, R_{MP} abruptly switched from $\sim 700 \Omega$ to $\sim 80 \Omega$ at about 700 mV, indicating the presence of only one threshold pH point (TP). **d.** Simulated number of TP as a function of the V_{OUTS} variation coefficient, confirming that V_{OUTS} variation of less than 1% ensures only one TP. The simulations in b-d were performed using 10 measured memristor devices, and the set-voltage variation of the devices was neglected to examine the pure effect of V_{OUTS} variation.

To highlight the advantages of the proposed memristor-based biosensor, we measured the retention characteristic of the fabricated memristors. The results showed that the memristor exhibits non-volatile memory with long-term retention of approximately 10 years at 431.3 K (see Figure R4), confirming its suitability for use in the electrochemical biosensor transducer with TS functionality. We further appended simulation results quantifying the effect of R_M variation on the transducer output voltage (V_{OUTS}), showing strong tolerance to R_M variation: a 19.2 % variation in HRS translates to less than 1 % variation in V_{OUTS} , thereby yielding only one TP (see Figure R5).

To address the reviewer's comment, we have appended Figures R4 and R5 and Table R2 as Supplementary Figures S14 and S7 and Supplementary Table S2 in the revised Supplementary Information, respectively. The corresponding explanations in the revised manuscript are as follows:

Page 17, line 291: **Although the memristors exhibited device-to-device resistance variation, the resistance parallelization scheme, by employing variation-tolerant R_{EQ} instead of directly**

using R_{MP} , effectively suppressed this variation and enabled a stable V_{OUTS} response with a clearly defined resistance switching characteristic of the parallel memristor cell (see Supplementary Figures S5 and S7).

Page 18, line 333: Our sensor technology is compared with previously reported semiconductor-based sensors (see Supplementary Table S2). To the best of our knowledge, the sensors developed in this work are the first demonstration of a semiconductor device-based electrochemical biosensor that exhibits TS functionality at the transducer level together with complete system-level integration. Unlike prior transducing devices, our memristor device also offers full CMOS compatibility, which is readily extendable to monolithic integration with SP blocks and thereby facilitating the realization of miniaturized sensor systems. Moreover, our transducer is the only one that simultaneously exhibits intrinsic TS functionality and long-term non-volatility (Supplementary Fig. S14), which is potentially valuable for miniaturized sensors with self-diagnosis capability requiring data retention.

Comment #1:

It is suggested that the authors propose a specific application scenario that is well-suited to pH threshold-biosensing capabilities.

Response: We sincerely appreciate the reviewer's constructive comments on our work and valuable suggestions for enhancing the quality of the study.

This study focuses on endowing the transducer with threshold-diagnosis capability and long-term storage of the diagnostic information. A typical sensor consists of sensing electrodes, a transducer, a signal-processing block, a communication block, and a computer or display. When the transducer itself has threshold-diagnosis and non-volatile retention capabilities, the subsequent signal-processing block can be considerably simplified through the elimination of ADCs, comparators, and MCUs. Moreover, because diagnosis is performed within the sensor, real-time external communication is no longer required, allowing removal of the communication block. Furthermore, because the diagnostic state is non-volatily stored in the transducer, the sensor needs power only during the diagnosis period; once the decision is recorded, power can be turned off without information loss, thereby drastically

reducing the required battery capacity.

Based on these characteristics, **our CMOS-compatible TS transducer technology with long-term non-volatile retention is highly suitable for the ultra-low-power operation and miniaturization of conventional electrochemical biosensors. A representative application is an ingestible sensor capsule** (see Figure R6). Conventional capsules¹⁴⁻¹⁶ rely on real-time wireless transmission to external receivers to diagnose diseases, forcing patients to wear a waist-mounted device. Their lengths typically exceed 2 cm, which can cause discomfort during swallowing and gastrointestinal transit. In contrast, our capsule eliminates real-time wireless transmission and external receivers by leveraging the transducer's intrinsic threshold-based diagnosis (see Figures R6a-R6c). The diagnosed information is stored non-volatily in the transducer, enabling retrieval after power is turned off and considerably decreasing the required battery capacity. Because batteries typically occupy more than 50 % of capsule volume, the overall form factor can be reduced to less than half of that of conventional systems, i.e., down to 11.5 mm in length and 5.5 mm in diameter. Power modeling indicates 13.2 days of operation at a 1-minute sampling interval and 4.7 days at a 20-second interval, assuming an 8.66 mA draw for 16.8 ms during each set operation of the simplified circuit (see Figure R6d).

The capsule can integrate multiple memristor-based transducers preset to the characteristic pH thresholds of digestive organs (stomach 2.6, small intestine 7.4, large intestine 6.5¹⁴). When the local pH drops below its threshold, the corresponding device switches from a high-resistance state (HRS) to a low-resistance state (LRS), effectively recording the physiological state in situ. The capsule typically exits the body within ~20 hours; once retrieved, it can be re-powered to read out the stored resistance states, enabling physicians to retrospectively identify abnormal conditions encountered along the gastrointestinal tract.

While memristor technology is rapidly advancing in biosensing applications, its specific use in gastrointestinal pH monitoring devices remains largely unexplored, indicating a valuable opportunity for innovation in future studies.

Figure R6. Potential application of the memristor-based electrochemical biosensor transducer with threshold-sensing functionality. **a.** Exploded view of memristor integrated ingestible sensor capsule. **b.** Schematic illustration of ingestible sensor capsule including multimodal sensors for gastrointestinal tract^{17,18}. **c.** Block diagram of the system. The ultra-low-leakage switch minimizes power loss during sleep mode. A nano-power system timer generates the clock signal for memristor operation and power switching, while a switch array manages the memristor array and its programming sequence. **d.** The estimated battery life equation and the estimated number of batteries when the measurement of the 20-second cycle is performed.

To address the reviewer’s comments, we have appended Figure R6 as Supplementary Figure S15 and in the revised Supplementary Information. The appended explanations in the revised manuscript are as follows:

Page 19, line 342: **Owing to its simplicity, scalability, and functionality, the proposed memristor-based technology holds strong potential for portable point-of-care diagnostic applications. Furthermore, in applications such as ingestible capsule sensors, this technology offers distinct advantages, such as miniaturization, non-volatile memory, scalability, and ultra-low power consumption, while addressing future challenges associated with safely monitoring internal physiological environments with minimal discomfort and risk to the user (Supplementary Fig. S15).**

Page 20, line 374: **Owing to its simplicity, scalability, and functionality, the proposed memristor-based technology holds strong potential not only for portable point-of-care**

diagnostic applications but also for miniaturized sensing devices that require inherent memory retention.

Comment #2:

Using only pH sensing as proof-of-concept fails to demonstrate relevance to actual diagnostic applications. Real biomarkers (glucose, troponin, etc.) in complex biological matrices are essential for meaningful validation.

Response: We sincerely thank the reviewer for the thorough evaluation of our work. As discussed in the previous comments, pH holds significant relevance for practical diagnostic applications such as gastrointestinal health monitoring, where changes in pH can indicate underlying conditions. However, as the reviewer rightly pointed out, demonstrating the utility of the memristor using only a pH sensor may appear less impactful in terms of its real-world applicability. To address this, we developed glucose and ascorbic acid (AA) sensors and investigated their electrocatalytic activities. Figure R9a shows the amperometric *i-t* response of the glucose sensor for increasing concentration of glucose from 0.25 mM to 2.0 mM at an applied potential of +0.4 V. For every addition of glucose, the sensor exhibited a rapid increase in catalytic current. The electrochemical stability of the developed sensor was further evaluated, showing excellent stability under dynamic conditions (Figure R9b). Similarly, the electrocatalytic activity of the AA sensor was examined under dynamic conditions with different concentrations of AA (25 – 500 μ M) and the obtained open circuit potential response is shown in Figure R9c. The electrochemical stability of the sensor was also investigated in the absence and presence of AA, demonstrating excellent stability (Figure R9d). These results confirm that the developed sensors exhibit excellent electrocatalytic activity and stability toward their respective target analytes.

Further, we have extended the utility of the memristor to the above developed glucose and AA sensors. When integrated with the memristor device, the sensor altered the device's resistance state upon reaching a predefined threshold. For demonstration purposes, we set thresholds of 1.25 mM for glucose and 100 μ M for AA. In the case of glucose, the memristor retained its high resistance state (HRS) when the glucose concentration was below the threshold. Upon reaching the threshold concentration, the memristor switched from a HRS to a low resistance state (LRS) (Supplementary Video 2). Conversely, for AA, the memristor switched its HRS to LRS when the concentration of AA was below threshold, and upon reaching the threshold

concentration, the memristor retained its HRS condition (Supplementary Video 3).

Figure R9. Electrochemical biosensing responses of glucose and ascorbic acid. **a.** Amperometric $i-t$ response of glucose sensor for the various concentrations of glucose (0.25 – 2.0 mM). For every addition of glucose, the sensor exhibited a rapid increase in catalytic current. **b.** The electrochemical stability of the developed sensor in the absence and presence of 1 mM glucose for 500 s, which showed excellent stability under dynamic conditions. Electrolyte: 0.1 M NaOH, Applied Potential: +0.4 V. **c.** The open circuit potential response of ascorbic acid (AA) sensor under dynamic conditions with different concentrations of AA (25 – 500 μ M) in 0.1 M PBS. **d.** The electrochemical stability of the sensor was investigated in the absence and presence of AA, demonstrating excellent stability for 500s. These results confirm that the developed sensors exhibit excellent electrocatalytic activity and stability toward their respective target analytes. Further, the utility of the memristor was investigated towards the electrochemical biosensing of glucose and AA. When integrated with the memristor device, the sensor altered the device's resistance state upon reaching a predefined threshold. For demonstration purposes, we set thresholds of 1.25 mM for glucose and 100 μ M for AA. In the case of glucose, the memristor retained its high resistance state (HRS) when the glucose concentration was below the threshold. Upon reaching the threshold concentration, the memristor switched from an HRS to a low resistance state (LRS) (Supplementary Video 2). Conversely, for AA, the memristor switched its HRS to LRS when the concentration of AA was below threshold, and upon reaching the threshold concentration, the memristor retained its HRS condition (Supplementary Video 3).

As per the reviewer's comment, the performance of the developed sensors was validated in complex biological matrices. For this purpose, the glucose sensor was investigated in diluted human serum samples, and the AA sensor in diluted human urine samples. Both the sensors demonstrated excellent catalytic activity even in the complex biological samples, which further supports the remarkable selectivity of the developed sensors. The obtained amperometric *i-t* and open circuit potential responses are shown in Figure R10.

To address the reviewer's comment, we have added Figures R9 and R10 as Supplementary Figures S10 and S12, and Supplementary Videos 2-3 in the revised Supplementary Information. The appended explanations in the revised manuscript are as follows:

Page 18, line 318: **Furthermore, to substantiate the applicability of the proposed memristor technology towards electrochemical biosensing, we extended its use to the detection of glucose and ascorbic acid (see Supplementary Figs. S10-S12 and Supplementary Video 2 & 3). The memristor device successfully switched its resistance state upon reaching the predefined threshold concentrations of these analytes, further validating the applicability of the proposed memristor-based biosensor to real biomarkers.**

Figure R10. Real-sample responses of glucose and ascorbic acid biosensors. (A) Amperometric *i-t* curve obtained for the glucose sensor for the addition of 2 mM glucose in 0.1 M NaOH containing 10% human serum at an applied potential of 0.4 V. (B) Open circuit potential response of AA sensor for the addition of 100 μM AA in 0.1 M PBS containing 5% human urine. The sensors exhibited stable performance in these complex biological matrices, confirming their suitability of the proposed memristor-based biosensor to real biomarkers.

Comment #3:

Device variability maybe the critical obstacle to practical applications. The stochastic nature of RRAM could lead to false positive result. Device-to-device variability characterization is insufficiently addressed in your work. Additionally, what quality control measures do you implement to ensure acceptable device-to-device consistency for reliable biosensor operation?

Response: We are grateful for the reviewer's insightful and constructive comments. As correctly pointed out, device-to-device (D2D) variability characterization was insufficiently addressed in the original manuscript. To address the reviewer's comments, we have included D2D variation characteristics and the associated quality-control measures. The detailed explanations and results are provided below.

1) Device-to-device variation characteristics

RRAM inherently exhibits resistance and set voltage variations. We evaluated memristors on the S5 die (hereafter, Raw memristors) and the same S5-die memristors after being mounted on a PCB for biosensor operation (hereafter, PCB memristors), as summarized in Figure R8. The variation coefficients of R_{HRS} , R_{LRS} , and V_{SET} were determined for both cases, and the PCB memristors exhibits relatively larger values compared with the Raw memristors (**R_{HRS} , R_{LRS} , and V_{SET} of Raw memristors = 15.2, 4.7, and 5.6 %**, respectively; **R_{HRS} , R_{LRS} , and V_{SET} of PCB memristors = 19.2, 8.6, and 6.7 %**, respectively) (see Figure R11). This degradation of the PCB memristors is mainly attributed to extrinsic factors, including wire bonding during board-level assembly and the absence of dedicated ESD protection circuitry in the present biosensor configuration. The degradation will be effectively mitigated through future monolithic CMOS integration of memristor-based transducers on a single chip, together with the incorporation of on-chip ESD protection schemes.

Figure R11. Measured variation characteristics of the fabricated S5 memristor devices. a. Device-to-device variation of the high-resistance state (R_{HRS}). **b.** Device-to-device variation of the low-resistance state (R_{LRS}). **c.** Device-to-device variation profile of the set voltage (V_{SET}). Both Raw memristors (from the S5 die) and PCB memristors (S5-die devices after PCB mounting) are shown together. The variation characteristics were extracted from 10 randomly selected devices. PCB memristors exhibited larger variation coefficients (R_{HRS} , R_{LRS} , and $V_{SET} = 19.2, 8.6,$ and 6.7%) than Raw memristors ($15.2, 4.7,$ and 5.6%), mainly due to extrinsic factors such as wire bonding and the absence of ES protection in the present biosensor configuration.

2) Quality control measures to effectively mitigate resistance variations in this work

To ensure that resistance variation does not degrade the diagnostic accuracy of the biosensor, we proposed a **novel transducer topology based on a resistance parallelization scheme** in this work. In this topology, the memristor resistance is connected in parallel with an integrated transistor, and the resulting parallel resistance (R_{EQ}) is connected to the output. By designing the transistor resistance (R_T) to be much smaller than the parallel memristor cell resistance (R_{MP}), R_{EQ} becomes highly insensitive to variations in R_{MP} , as dictated by the parallel resistance relation ($R_1/R_2 = (R_1 \times R_2)/(R_1 + R_2)$). This stabilized R_{EQ} is multiplied by I_{DS} to generate the output voltage in the sensing stage (V_{OUTS}) (see Figure R5a). Figures R5b-R5d

present corresponding LTspice simulation results of the biosensor transducer. The I–V characteristics of the modeled memristors were well fitted to those of the measured PCB memristors (see Figure R2a and revised Supplementary Figure S6a). The R_{HRS} variation coefficient of 19.2 % obtained in this work was **considerably reduced to less than 1% for R_{EQ}** , thereby yielding a memristor resistance–tolerant V_{OUTS} with a variation coefficient **below 1%** (see Figure R5b). This stable characteristic of V_{OUTS} exhibited a clear HRS-to-LRS transition at a single V_{OUTS} step, indicating that the **biosensor transducer possesses only one threshold pH point** (see Figures R5c–R5d). This variation-immune V_{OUTS} enabled the implemented pH sensor system to exhibit the same threshold pH point within a one-pH interval (Fig. 6d in the revised manuscript).

Figure R5. The effect of the resistance parallelization scheme on the biosensor transducer. a. Circuit diagram of the biosensor transducer at the sensing stage. **b.** Simulated parallel resistance (R_{EQ}) and sensing-stage output voltage (V_{OUTS}) characteristics based on the measured R_{HRS} (HRS resistance of memristors) variation of 19.2%. The R_{EQ} variation coefficient was considerably reduced to less than

1%, resulting in a V_{OUTS} variation coefficient also below 1%. **c.** Simulated switching characteristics of the parallel memristor cell resistance (R_{MP}) under increasing V_{OUTS} with a variation coefficient of less than 1%. With V_{OUTS} increased in ~ 26 mV step, R_{MP} abruptly switched from $\sim 700 \Omega$ to $\sim 80 \Omega$ at about 700 mV, indicating the presence of only one threshold pH point (TP). **d.** Simulated number of TP as a function of the V_{OUTS} variation coefficient, confirming that V_{OUTS} variation of less than 1% ensures only one TP. The simulations in b-d were performed using 10 measured memristor devices, and the set-voltage variation of the devices was neglected to examine the pure effect of V_{OUTS} variation.

3) Quality control measures to effectively mitigate V_{SET} variations as a future plan

This work was conducted as a proof-of-concept study for the proposed biosensor transducer, and therefore measures to address set voltage (V_{SET}) variation of the memristors were not included within the present scope. Effective control of V_{SET} variation is critical for ensuring reliable biosensor operation. To clarify our future directions, we have added a discussion describing three possible approaches for mitigating V_{SET} variation, as summarized below:

Approach 1) Application of filament-confinement memristors with low V_{SET} variation. In this work, the D2D variation coefficient of ten PCB-mounted memristor devices was measured to be as low as 6.7% (see Figures R11c and R12a), which we attribute to the use of a stable 180 nm CMOS BEOL process and the low Gibbs free-energy-based redox reactions of the $\text{Ta}_2\text{O}_5/\text{TaO}_x$ switching layer. Recent studies have demonstrated remarkable progress, with CMOS-incompatible memristors exhibiting **V_{SET} variation coefficients as low as $\sim 4\%$** ²¹. In this context, we have already been granted a patent for a reproducible filament-confinement technique in a CMOS-compatible process²², which is currently under development. Upon its completion, we will report the detailed results in a separate journal publication.

Approach 2) Optimization of transducer circuit design. In this work, the measured ΔV_w was 57 mV/pH, which is comparable to previous results (see Figure 6d in the main manuscript). However, this ΔV_w of 57 mV was reduced to a ΔV_{OUTS} of ~ 26 mV after passing through a series-connected transistor pair shown in Fig. 5a. For achieving high diagnostic accuracy of the biosensor, it is essential to secure a wide ΔV_{OUTS} that can encompass V_{SET} variation. In this work, the gate width ratio between the upper transistor and the lower transistor in the series-connected transistor pair was set to 1. By increasing this gate width ratio to more than 10, together with gate-bias optimization of the transistors, **a ΔV_{OUTS} exceeding 50 mV was obtained in our transducer simulation** (see Figure R12b). This improvement is attributed to the enhanced trans-conductance (g_m) of the upper transistor due to its increased geometrical gate width, which represents a general device-level design technique to improve g_m .

Approach 3) System-level algorithm for mitigating memristor variability. V_{SET} variation of memristor ReRAM devices generally follows a Gaussian distribution, and devices located in the distribution tail degrade the diagnostic accuracy of the biosensor. To address this, we will apply a system-level algorithm that will enable reliable biosensor operation by utilizing only devices with V_{SET} distribution confined within $\pm 1\sigma$. This confinement will be realized through “sensor trimming”, which will compensate for **differences in the average V_{SET} values among devices** by adjusting the V_{REF} voltage (as already demonstrated in Figure 6e of the main manuscript). **Intra-device V_{SET} variation** will further be managed by a “**repeated-measurement and averaging**” algorithm, in which each measurement is repeated more than ten times, the results are averaged, and outliers with large deviations are discarded.

When the three approaches are applied, namely, confining the memristor V_{SET} variation to below 4%, widening ΔV_{OUTS} to greater than 50 mV, and applying a system-level algorithm that effectively utilizes only devices within $\pm 1\sigma$, our biosensor is expected to achieve error-free threshold detection within an interval of one pH unit (see Figures R12c and R12d).

To address the reviewer’s comment, we have appended Figures R11, R5, and R12 as Supplementary Figures S5, S7, and S16 in the revised Supplementary Information, respectively. The corresponding explanations have been appended in Supplementary Information as Supplementary Note S1. The appended explanations in the revised manuscript are as follows:

Page 15, line 257: The variation factor (σ/μ) of resistance of the S5 devices reached approximately 20 %, as shown in Figs. 3b and 4c and **Supplementary Fig. S5**.

Page 17, line 291: **Although the memristors exhibited device-to-device resistance variation, the resistance parallelization scheme, by employing variation-tolerant R_{EQ} instead of directly using R_{MP} , effectively suppressed this variation and enabled a stable V_{OUTS} response with a clearly defined resistance switching characteristic of the parallel memristor cell (see Supplementary Figures S5 and S7).**

Page 19, line 348: **Although this study was conducted as a proof-of-concept for the proposed biosensor transducer, effective control of V_{SET} variation of the memristors is critical for ensuring reliable biosensor operation. Possible mitigation strategies at the device, circuit, and system levels are summarized in Supplementary Fig. S16 and Supplementary Note 1.**

Figure R12. Strategy for biosensor quality control against memristor set voltage variation. **a.** Measured set voltage (V_{SET}) variation for ten PCB-mounted memristor devices. The D2D variation coefficient of ten PCB-mounted memristor devices was measured to be 6.7% **b.** Optimization of transducer circuit design by adjusting the gate width ratio of the series-connected transistor pair and the gate bias (V_{GS}) of the lower transistor. The gate width ratio is defined as the geometrical gate width of the upper transistor divided by that of the lower transistor, as shown in Fig. 5a. An increase in the gate width ratio increased ΔV_{OUTS} from 26 mV to 57 mV. **c.** Simulated operation of the threshold-sensing (TS) biosensor transducer with three elements for effectively reducing V_{SET} variation: a memristor V_{SET} variation coefficient of 4%, a ΔV_{OUTS} exceeding 50 mV, a memristor V_{SET} variation coefficient of less than 4%, a ΔV_{OUTS} exceeding 50 mV, and the use of only devices within the $\pm 1\sigma$ range of the V_{SET} distribution, representing the effect of a system-level algorithm. Parallel memristor resistance (R_{MP}) switched from HRS to LRS at only the single step of about 700 mV, indicating the presence of only one threshold pH point (TP). The detailed explanations regarding the three elements for mitigating V_{SET} variation are presented in Supplementary Note S1. **d.** Simulated number of TP of the biosensor transducer as a function of the memristor V_{SET} variation coefficient under the same conditions as in **c**, indicating that memristor V_{SET} variation coefficient below 4% enables the biosensor to achieve error-free threshold detection. The simulation was performed with 200 memristor devices.

Comment #4:

Clinical samples contain numerous potential interferents that could compromise specificity. Interference testing is needed.

Response: We fully agree that clinical samples contain a complex matrix of potential interferents that may affect biosensor specificity. To address this, we conducted interference testing by evaluating the responses of both the glucose and ascorbic acid sensors in the presence of common interfering species. Figure R13a portrays the amperometric *i-t* curve of the glucose sensor in the presence of various interferents, including urea, uric acid, cholesterol, NaCl, KCl, nitrite, ascorbic acid, hydrogen peroxide and dopamine in 0.1 M NaOH at +0.4 V. Similarly, Figure R13b displays the open circuit potential curve obtained for the AA sensor in the presence of different interfering species such as uric acid, urea, glucose, dopamine, cholesterol and hydrogen peroxide in 0.1 M PBS. These results demonstrate that both sensors maintained high selectivity under the tested conditions. Furthermore, we evaluated the practical applicability of the sensors using real biological samples, including diluted human serum and urine (Figure R10). The sensors exhibited stable performance in these complex matrices, confirming the applicability of the proposed memristor-based biosensor to real biomarkers.

Figure R13. Selectivity response of glucose and ascorbic acid biosensors. a. Amperometric *i-t* curve of the glucose sensor in the presence of various interferents, including urea, uric acid, cholesterol, NaCl, KCl, nitrite, ascorbic acid, hydrogen peroxide and dopamine in 0.1 M NaOH at +0.4 V. **b.** Open circuit potential curve obtained for the AA sensor in the presence of different interfering species such as uric acid, urea, glucose, dopamine, cholesterol and hydrogen peroxide in 0.1 M PBS. These results demonstrate that both sensors maintained high selectivity under the tested conditions.

To address the reviewer's comment, we have appended Figure R13 as Supplementary Figure S11 in the revised Supplementary Information, respectively. The appended explanations in the revised manuscript are as follows:

Page 18, line 318: **Furthermore, to substantiate the applicability of the proposed memristor technology towards electrochemical biosensing, we extended its use to the detection of glucose and ascorbic acid (see Supplementary Figs. S10-S12 and Supplementary Video 2 & 3). The memristor device successfully switched its resistance state upon reaching the predefined threshold concentrations of these analytes, further validating the applicability of the proposed memristor-based biosensor to real biomarkers.**

Comment #5:

Error bars are missing in Fig 6. Were these measurements obtained from a single device, or do they represent averaged data from multiple independent devices?

Response: We sincerely appreciate the reviewer's valuable comment. According to your suggestion, we have added error bars in Figure 6. Figures 6a–6d show the data obtained from three independent devices, as shown in Figure R14, while Figures 6e–6f show the data obtained from a single device in the revised manuscript.

This study was conducted as a proof-of-concept to demonstrate the feasibility of the proposed memristor-based threshold-sensing (TS) biosensor. To the best of our knowledge, a transducer capable of performing threshold sensing at the system level using memristors has never been reported before. By introducing a resistance-parallelization scheme that effectively suppresses memristor resistance variation, we achieved an identical threshold pH point (TP) for the implemented biosensor with three different biosensors, each employing distinct parallel memristor cells.

However, the TP might differ if a larger number of devices are used. To analyze the impact of device variation on the diagnostic accuracy of the biosensor, we further performed circuit-level simulations in LTspice using 200 modeled memristor devices accurately fitted to experimental data. With a 4% V_{SET} variation and $\Delta V_{OUTS} > 50$ mV, the memristor-based

biosensor is expected to be practically applicable as a pH sensor when combined with a system-level algorithm (see our response to Comment #4, together with Figures R11, R5, R12, S5, S7, S16, and Supplementary Note 1). Furthermore, as our next study, we plan to extend this work by effectively minimizing device variation of the memristor, and the results will be reported once available.

Figure R14. Comparison of Figure 6 before (top) and after (bottom) the insertion of error bars

To address the reviewer's comment, we have updated Figures 6a, 6c, and 6d in the revised manuscript. The appended explanations in the revised manuscript are as follows:

Page 16, lines 12 in Fig. 6 caption: **Experimental repetitions: a, c, and d were obtained from different memristor cells (n=3), and e and f from a memristor cell (n=1).**

Comment #6:

Will temperature and humidity affect the stability of the sensor? What environmental compensation strategies will you implement?

Response: We sincerely appreciate the reviewer's valuable comment. Yes, temperature and humidity can influence the performance and long-term stability of electrochemical sensors, as they affect reaction kinetics, material properties, and signal baseline. In our current work, the primary focus is placed on the experimental validation of the proposed memristor-based transducer: therefore, environmental compensation strategies are not the main emphasis of this work. The sensor was operated at room temperature, and no significant changes were observed due to humidity, as all measurements were conducted in aqueous solutions.

We fully acknowledge that this is a key factor in real-world applications. It is well known that all semiconductor devices exhibit performance variations depending on temperature and humidity, and RRAM is no exception. To tackle this issue, we are exploring strategies to compensate for environmental factors. These include integrating temperature and humidity sensors for real-time calibration, and well as encapsulating the sensing element with breathable yet protective coatings. Such approaches are expected to enhance sensor durability and maintain accuracy under varying environmental conditions.

Should the reviewer consider additional information or measurements on these points helpful, we will be glad to incorporate them promptly.

References:

1. Tzouvadaki, I., Lu, X., Micheli, G. D., Ingebrandt, S., and Carrara, S., Nano-fabricated memristive biosensors for biomedical applications with liquid and dried samples, *2016 38th Annual International Conference of the IEEE Engineering in Medicine and Biology Society (EMBC)*, 2016.
2. Janissen, R., Sahoo, P., Santos, C., Silva, A., Zuben, A., Souto, D., Costa, A., Celedon, Zanchin, P., N., Almeida, D., Oliveira, D., Kubota, L., Cesar, C., Souza, A., and Cotta, M., InP Nanowire Biosensor with Tailored Biofunctionalization: Ultrasensitive and Highly Selective Disease Biomarker Detection, *ACS Nano Lett.*, 2017, **17**, 5938-5949.
3. Lei, Y. M., Xiao, M. M., Li, Y. T., Xu, L., Zhang, H., Zhang, Z. Y., and Zhang, G., Detection of heart failure-related biomarker in whole blood with graphene field effect transistor biosensor, *Biosens Bioelectron.*, 2017, **91**, 1-7.
4. Kwon, J., Lee, Y., Lee, T., and Ahn, J., Aptamer-Based Field-Effect Transistor for Detection of Avian Influenza Virus in Chicken Serum, *ACS Anal. Chem.*, 2020, **92**, 5524–5531.
5. Tzouvadaki, I., Stathopoulos, S., Abbey, T., Michalas, L., and Prodromakis, T., Monitoring PSA levels as chemical state-variables in metal-oxide memristors, *Sci. Rep.*, 2020, **10**, 15281(1-6).
6. Kumar, P., Maikap, S., Ginnaram, S., Qiu, J., Jana, D., Chakrabarti, S., Samanta, S., Singh, K., Roy, A., Jana, S., Dutta, M., Chang, Y., Cheng, H., Mahapatra, R., Chiu, H., and Yang, J., Cross-Point Resistive Switching Memory and Urea Sensing by Using Annealed GdOx Film in IrOx/GdOx/W Structure for Biomedical Applications, *J. Electrochem. Soc.*, 2017, **164**, B127-B135.
7. Knapic, D. et al. Anodic Niobium–Titanium Oxide Crossbar Memristor Arrays for pH Sensing in Liquids. *Phys. Status Solidi A.*, 221, 2300878 (2024).
8. Sinhaa, S., Pal, T., Kumara, D., Sharma, R., Kharbanda, D., Khanna, P. K., Mukhiya, R., Design, fabrication and characterization of TiN sensing film-based ISFET pH sensor, *Materials Letters*, 2021, **304**, 130556.
9. Kim, H., Kim, J., Jeong, M., Lee, D., Kim, J., Lee, M., Kim, G., Kim, J., Lee, J., and Lee, J., Bioelectronic Sutures with Electrochemical pH-Sensing for Long-Term Monitoring of the Wound Healing Progress, *Advanced Functional Materials*, 2024, **34**, 2402501.
10. Zhou, G. Full hardware implementation of neuromorphic visual system based on multimodal optoelectronic resistive memory arrays for versatile image processing. *Nat. Commun.*, **14**, 8489 (2023).

11. Doremaele, E.R.W. et al. A retrainable neuromorphic biosensor for on-chip learning and classification. *Nat. Electron.*, **6**, 765-770 (2023).
12. Liu, D. et al. A wearable in-sensor computing platform based on stretchable organic electrochemical transistors. *Nat. Electron.*, **7**, 1176-1185 (2024).
13. Ji, J. et al. Single-transistor organic electrochemical neurons. *Nat. Commun.*, **16**, 4334 (2025).
14. Even, A. et al. Measurements of redox balance along the gut using a miniaturized ingestible sensor. *Nat. Electron.*, **8**, (2025).
15. Wang, Y.C., Pan, J., Liu, Y.W., Sun, F.Y., Qian, Y.Y., Jiang, X., Zou, W.B., Xia, J., Jiang, B., Ru, N., Zhu, J.H., Linghu, E.Q., Li, Z.S., Liao, Z. Adverse events of video capsule endoscopy over the past two decades: a systematic review and proportion meta-analysis. *BMC Gastroenterol.* **20**, 364 (2020), doi: 10.1186/s12876-020-01491-w.
16. Gounella, R. et al. Endoscope capsules: The present situation and future outlooks, *Bioengineering*, **10**, 1347 (2023).
17. Holt, B.M., Stine, J.M., Beardslee, L.A., Ayansola, H., Jin, Y., Pasricha, P.J., and Ghodssi, R., An ingestible bioimpedance sensing device for wireless monitoring of epithelial barriers. *Microsyst Nanoeng.* **11**, 24. (2025).
18. Even, A., Minderhoud, R., Torfs, T., Leonardi, F., van Heusden, A., Sijabat, R., Firfilionis, D., Castro Miller, I. D., Rammouz, R., Teichmann, T., van Bergen, R., Vermeeren, G., Capuano, E., Armstrong, R., Mathwig, K., de Vries, S., Goris, A., Van Helleputte, N., Hooiveld, G., Van Hoof, C. Measurements of Redox Balance along the Gut Using a Miniaturized Ingestible Sensor. *Nat. Electron.* (2025).
19. Oh, J.-S. et al. Structure and Formation Mechanisms in Tantalum and Niobium Oxides in Superconducting Quantum Circuits, *ACS Nano*, **18**, 19732-19741 (2024).
20. Park, et al., In situ observation of filamentary conducting channels in an asymmetric Ta₂O_{5-x}/TaO_{2-x} bilayer structure, *Nat. Commun.*, **4**, 2382 (2013)
21. Sun, D., Zhu, X., Chen, S., Fang, H., Zhu, G., Lan, G., He, L., and Shi, Y. Uniformity, Linearity, and Symmetry Enhancement in TiO_x/MoS_{2-x}O_x Based Analog RRAM via S-Vacancy Confined Nanofilament. *Nano Letters*, **24**, 16283-16292 (2024)
22. Lee, J., Lee, W.-C. & Cho, H.-J. Resistive Memory Device With 1T-1R and The Fabrication Method Of The Same, Korean Patent 10-2820732-0000, (2025). Available at: <https://www.kipris.or.kr/khome/search/searchResult.do>

Response Letter to Reviewers' Comments

We sincerely appreciate the reviewers' time and effort in evaluating our manuscript and for providing additional comments and suggestions that have helped improve the quality of our work. In response to the reviewers' evaluations, we have made a point-by-point response to the reviewers' comments. All modifications and additions in the revised manuscript are highlighted in red. Our detailed point-by-point responses to the reviewers' comments are as follows.

Reviewer #3

The authors have addressed most of my comments. The revised manuscript is much clearer in its novelty. Just a minor point: Although this work was conducted as a proof-of-concept study for the proposed biosensor transducer, detecting glucose and AA in a strong alkaline environment is not suitable for physiological conditions.

Response: We sincerely appreciate the reviewer's constructive comments on our work and valuable suggestions for enhancing the quality of the study. To address the reviewer's comment, we have appended the revised manuscript and revised Supplementary Information as follows:

Page 12, line 310:

- Previous manuscript: ~, we extended its use to the detection of glucose and ascorbic acid (see Supplementary Figs. S10-S12 and Supplementary Video 2 & 3).
- Revised manuscript: ~, we extended its use to the detection of glucose and ascorbic acid **with appropriate sample preprocessing** (see Supplementary Figs. S10-S12 and Supplementary Video 2 & 3).

Page 12, line 313:

- Previous manuscript: ~, further validating the applicability of the proposed memristor-based biosensor to real biomarkers.
- Revised manuscript: ~, further validating the **demonstration** of the proposed memristor-based biosensor **with** real biomarkers.

Page 13, line 333:

- Previous manuscript: Owing to its simplicity, scalability, and functionality, the proposed memristor-based technology holds strong potential for portable point-of-care diagnostic applications. Furthermore, in applications such as ingestible capsule sensors, this technology offers distinct advantages, such as miniaturization, non-volatile memory, scalability, and ultra-low power consumption, while addressing future challenges associated with safely monitoring internal physiological environments with minimal discomfort and risk to the user (Supplementary Fig. S15).

- Revised manuscript: Owing to its simplicity, scalability, and functionality, the proposed memristor-based technology **shows promise for future integration into portable sensing platforms, including point-of-care and health-monitoring systems.** Furthermore, in **miniaturized electronic devices** such as ingestible capsule **prototypes**, this **approach could** offer advantages, such as miniaturization, non-volatile memory, scalability, and ultra-low power consumption, **potentially contributing to future efforts aimed at** monitoring internal physiological environments **in a safe and minimally invasive manner** (Supplementary Fig. S15).

Page 13, line 343:

- Revised manuscript: **Furthermore, further efforts are required to validate the sensing functionality under realistic biochemical conditions. The proposed memristor-based transducer was evaluated for biosensing applications using pH, glucose and ascorbic acid, as representative analytes. Among these, glucose detection currently operates effectively only under strong alkaline conditions, indicating the need for further material and interface optimization. Future studies should therefore assess device performance in complex biofluid environments to understand the influence of factors such as biofouling, signal drift, and cross-sensitivity. Ultimately, comprehensive validation with clinically relevant samples under real-world operating conditions will be essential to establish the broader translational potential of this technology.**

Page 14, line 370:

- Previous manuscript: ~ memristor-based technology holds strong potential not only for portable point-of-care diagnostic applications but also for miniaturized sensing devices that require inherent memory retention.

- Revised manuscript: ~ memristor-based technology **shows promise for future integration**

into portable sensing platforms and miniaturized devices that could benefit from inherent memory retention.

Page 11, line 240:

- Previous Supplementary Information: ~ matrices, confirming their suitability of the proposed memristor-based biosensor to real biomarkers.
- Revised Supplementary Information: ~ matrices, **demonstrating the** suitability of the proposed memristor-based biosensor **for detecting relevant biomolecules**.

Page 13, line 281:

- Previous Supplementary Information: Supplementary Figure S15. Potential application of the memristor-based
- Revised Supplementary Information: Supplementary Figure S15. Potential application **and future outlook** of the memristor-based